# Impaired upper respiratory tract barrier function during postnatal development predisposes to invasive pneumococcal disease

Kristen L. Lokken-Toyli[1]*, Surya D. Aggarwal[1], Gavyn Chern Wei Bee[1], Wouter A. A. de Steenhuijsen Piters[2,3], Cindy Wu[1], Kenny Zhi Ming Chen[1], Cynthia Loomis[4], Debby Bogaert[5,6], Jeffrey N. Weiser[1]

1 Department of Microbiology, New York University School of Medicine, New York, New York, United States of America, 2 Department of Paediatric Immunology and Infectious Diseases, Wilhelmina Children's Hospital/University Medical Center Utrecht, Utrecht, the Netherlands, 3 National Institute for Public Health and the Environment, Bilthoven, the Netherlands; Department of Parasitology, Leiden University Medical Center, Leiden, the Netherlands, 4 Department of Pathology, New York University School of Medicine, New York, New York, United States of America, 5 Department of Paediatric Immunology and Infectious Diseases, Wilhelmina Children's Hospital/University Medical Center Utrecht, Utrecht, the Netherlands, 6 Centre for Inflammation Research, Institute for Regeneration and Repair, Queen's Medical Research Institute, University of Edinburgh, Edinburgh, United Kingdom

* Kristen.Lokken-toyli@nyulangone.org

**Data Availability Statement:** Mouse RNA-sequencing data has been uploaded to the GEO database under GE accession number GSE116604.

## Abstract

Infants are highly susceptible to invasive respiratory and gastrointestinal infections. To elucidate the age-dependent mechanism(s) that drive bacterial spread from the mucosa, we developed an infant mouse model using the prevalent pediatric respiratory pathogen, *Streptococcus pneumoniae* (*Spn*). Despite similar upper respiratory tract (URT) colonization levels, the survival rate of *Spn*-infected infant mice was significantly decreased compared to adults and corresponded with *Spn* dissemination to the bloodstream. An increased rate of pneumococcal bacteremia in early life beyond the newborn period was attributed to increased bacterial translocation across the URT barrier. Bacterial dissemination in infant mice was independent of URT monocyte or neutrophil infiltration, phagocyte-derived ROS or RNS, inflammation mediated by toll-like receptor 2 or interleukin 1 receptor signaling, or the pore-forming toxin pneumolysin. Using molecular barcoding of *Spn*, we found that only a minority of bacterial clones in the nasopharynx disseminated to the blood in infant mice, indicating the absence of robust URT barrier breakdown. Rather, transcriptional profiling of the URT epithelium revealed a failure of infant mice to upregulate genes involved in the tight junction pathway. Expression of many such genes was also decreased in early life in humans. Infant mice also showed increased URT barrier permeability and delayed mucociliary clearance during the first two weeks of life, which corresponded with tighter attachment of bacteria to the respiratory epithelium. Together, these results demonstrate a window of vulnerability during postnatal development when altered mucosal barrier function facilitates bacterial dissemination.

Human gene expression data have been deposited at the National Centre for Biotechnology Information GenBank database under accession no. GSE152951. Full patient metadata are available upon request from the Spaarne hospital at WetenschapsBureau@spaarnegasthuis.nl. All other relevant data are within the manuscript and Supporting Information files.

**Funding:** This work was supported by the following grants, Division of Intramural Research, National Institute of Allergy and Infectious Diseases, 1AI150893 to JNW; Division of Intramural Research, National Institute of Allergy and Infectious Diseases, AI038446 to JNW; Division of Intramural Research, National Institute of Allergy and Infectious Diseases, AI143043 to KLLT; Division of Intramural Research, National Institute of Allergy and Infectious Diseases, A1007180-35 to KLLT. The funders had no role in study design, data collection and analysis, decision to publish or preparation of the manuscript. No authors received a salary from any funders.

**Competing interests:** The authors have declared that no competing interests exist.

## Author summary

In children, bacterial respiratory and gastrointestinal pathogens remain major contributors to childhood mortality worldwide. Yet, the immune deficiencies intrinsic to early age that compromise bacterial control during infection remain relatively unknown. In this study, we utilized clinically relevant infant mouse models and a human newborn study to characterize mucosal barrier defenses that are critical for mediating resistance to invasive infection. Using the prominent pediatric respiratory pathogen, *Streptococcus pneumoniae* (*Spn*), we show that the postnatal period is associated with an increased susceptibility to invasive *Spn* infection following nasal colonization. In contrast to adult mice, infant mice exhibited increased bacterial translocation across the nasal mucosal barrier that was independent of robust barrier damage. Instead, transcriptional analysis of the nasal epithelium in infant mice revealed decreased expression of tight junction proteins that corresponded with increased barrier permeability and delayed mucociliary clearance. Strikingly, a similar transcriptional signature was also observed in nasopharyngeal samples collected from human infants during the first year of life. These findings suggested that disruption in barrier integrity and function may be a hallmark of postnatal development. Collectively, our data underscores a vulnerable period during early life when impaired nasal barrier defenses facilitate bacterial translocation, culminating in invasive infection.

## Introduction

Severe respiratory and gastrointestinal infections consistently remain leading causes of mortality in children under 5 years old [1]. Despite epidemiological evidence indicating young age as a significant risk factor for invasive infection, the precise age-dependent mechanism(s) driving susceptibility remain mostly unknown [2,3]. The process of bacterial invasion begins with stable colonization of URT, which is mediated by intimate attachment to epithelial cells, and is followed by translocation across the epithelium, migration through the extracellular matrix, and crossing the endothelium into the bloodstream. As the first line of defense, the mucosal epithelium plays a substantial role in the initial safeguarding of the host from infectious disease. However, a clear understanding of the mechanism(s) utilized by prominent respiratory pathogens to breach this initial barrier in the context of early life is lacking.

The mucosal epithelium of the respiratory tract is composed of a heterogeneous cell population that serves as a physical barrier to prevent the passage of foreign particles and pathogens [4,5]. In order to limit bacterial adhesion to the epithelial surface, the mucosal barrier produces copious amounts of mucus, antimicrobial peptides, and immunoglobulins [6,7]. Additionally, epithelial cell turnover and mechanical clearing of bacteria through mucociliary clearance prevents persistent bacterial colonization [8]. The structural and functional integrity of the mucosal epithelium plays a key role in restricting paracellular passage of pathogens. Cell-cell junctions are polarized and sealed by a dense protein network consisting of tight junctions, adherence junctions, gap junctions, and desmosomes, which prevent passage of small and large molecules, including water, ions, and proteins [9]. Beyond serving as a physical barrier, the mucosal epithelium orchestrates the innate and adaptive immune responses to viruses and bacteria. Sensing of potential pathogens by epithelial cells initiates inflammatory cascades that are necessary for efficient bacterial clearance, but these cascades must be selectively induced to prevent disruption of epithelial barrier integrity during homeostasis [5].

As a frequent colonizer of the upper respiratory tract (URT), carriage of *Streptococcus pneumoniae* (*Spn*) is typically asymptomatic; however, in young children and the elderly, colonization can progress to invasive pneumococcal disease (IPD) [2]. The etiology of *Spn* bloodstream infections differs between infants and adults. While invasive *Spn* infections often stem from pneumonia in adults [10], young children frequently exhibit occult bacteremia following nasopharyngeal colonization, indicating direct seeding to the bloodstream [11]. The precise mechanism utilized by *Spn* for crossing the nasal mucosal barrier *in vivo* remains unclear. Previous work using adult murine models and *in vitro* cell lines implicated both paracellular migration and receptor-mediated endocytosis as mechanisms by which *Spn* can use to traverse the nasal epithelium [12,13]. Additionally, data from an experimental human pneumococcal carriage model showed that *Spn* forms micro-colonies and crosses the epithelium either by endocytosis or paracellular movement in the absence of overt barrier damage or inflammation [14]. The mechanism utilized by *Spn* to disseminate from the URT to the bloodstream during early life is unknown. Direct access to the bloodstream by colonizing microbes is also commonly observed with other pediatric pathogens capable of surviving in the blood following invasion, including *Streptococcus*, *Haemophilus*, *Neisseria*, and *Salmonella* species [3]. Despite this conserved feature among many pediatric pathogens, the precise mechanisms behind transition from colonization to invasive disease during young age is not well understood. Here, we utilized *Spn* as a model pathogen afflicting young children to investigate how age compromises control to invasive infection.

## Results

### Young age increases susceptibility to invasive *Spn* infection

To determine how age impacts susceptibility to invasive pneumococcal infection, we developed an infant infection model using a mouse virulent type 6A strain of *Streptococcus pneumoniae* (*Spn* T6A). Infant (10 day-old) and adult (8–10 week-old) mice were infected with either $1 \times 10^3$ or $1 \times 10^5$ CFU, respectively, via the intranasal (IN) route in the absence of anesthesia to prevent aspiration of bacteria into the lower respiratory tract. After URT infection, we observed a significant reduction in survival of infant mice (38%) compared to adult mice (93%) at 14 days post-infection (dpi) (**Fig 1A**). Adult mice remained resistant to developing pneumococcal bacteremia despite being inoculated IN with 10- and 100-fold more bacteria (**S1A Fig**). Consistent with septic death, infant mice had significantly higher blood (**Fig 1B**) and spleen (**Fig 1C**) bacterial loads at 3 dpi, despite similar levels of URT colonization compared to adults at 3 dpi (**Fig 1D**). Our results demonstrate that young age is a risk factor for invasive *Spn* infection from URT colonization, consistent with observations in humans.

### Increased *Spn* translocation across the URT barrier underlies early life susceptibility to pneumococcal sepsis

The increased rate of pneumococcal bacteremia during URT colonization in infant mice suggested two possibilities that facilitate bacterial dissemination, differences in systemic clearance and/or URT barrier function. To assess the effect of age on systemic control of *Spn*, we infected infant and adult mice intraperitoneally (IP) with $1 \times 10^2$ CFU of *Spn* T6A, to bypass the epithelial barrier, and determined bacterial load 1 dpi. Although we observed age-related differences in bacterial control once beyond the barrier, both infant and adult mice were highly susceptible to systemic *Spn* infection in the blood and spleen (**Figs 2A and S2A**).

Next, we investigated bacterial and host factors known to affect bloodstream infection. One important bacterial factor is the capsule, which minimizes complement deposition and,

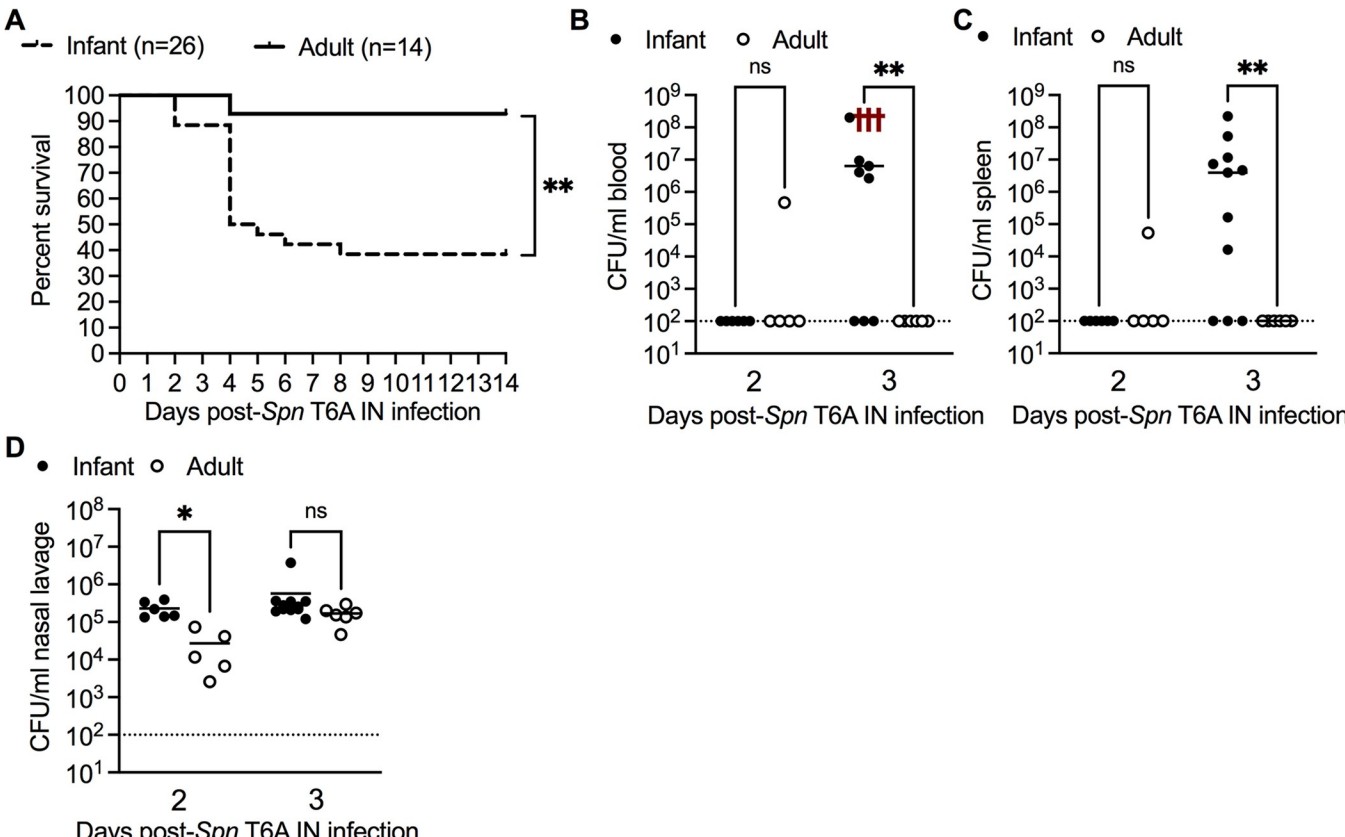

**Fig 1. Young age increases susceptibility to invasive pneumococcal infection. A,** Percent survival of infant (n = 26) and adult (n = 14) mice at 14 days post intranasal (IN) infection with $10^3$ or $10^5$ CFU, respectively, of *Streptococcus pneumoniae* type 6A (*Spn* T6A). Data are from 3 independent experiments. Statistical significance determined using Log-rank Mantel-Cox test. **, $p \leq 0.01$. **B,** Colony-forming units (CFU) of *Spn* T6A in blood, (**C**) spleen and (**D**) nasal lavage from infant and adult mice at 2 and 3 days post infection (dpi) (n = 6–13). Data represent individual mice with the median and are collected from 1–2 experiment(s). Statistical significance determined using One-way ANOVA with a Kruskal-wallis post-hoc multiple comparisons test. *, $p \leq 0.05$; **, $p \leq 0.01$; ns, not significant. Red crosses in the blood CFU figure indicate non-surviving mice and were excluded from the statistical analysis. Dotted line represents limit of detection.

subsequently, reduces phagocyte-mediated clearance [15]. Capsule also promotes intracellular survival of *Spn* within vascular endothelial cells, as well as the translocation of *Spn* across these cells [16]. To further assess the ability of infant mice to control systemic *Spn* infection, we compared survival in the bloodstream of isogenic strains with a single amino acid change in CpsE that led to a 2.4-fold difference in amount of type 6A capsular polysaccharide (**S2B Fig**). This mutation was associated with an inability to cause sustained bacteremia after IP administration (**Fig 2B**), or septic death following IN colonization (**Fig 2C**), despite similar nasal colonization levels at 14 dpi (**S2C Fig**). These results showed that survival of *Spn* in the bloodstream was largely dependent on important bacterial factors, rather than age-dependent differences in mechanisms of bacterial clearance.

Therefore, we hypothesized that protection against *Spn* bacteremia during URT colonization in adult mice, in contrast to infants, relied on the respiratory barrier restricting the migration of *Spn* from the URT to the bloodstream. To determine whether age impacts *Spn* translocation across the barrier, we IP treated infant and adult mice with cobra venom factor (CoVF), to deplete complement factor 3 (C3), or dPBS, and IN infected with *Spn*. Treatment with CoVF was previously shown to render animals more susceptible to bloodstream infection [17,18], which allowed us to lower the bacterial threshold required to sustain bacteremia.

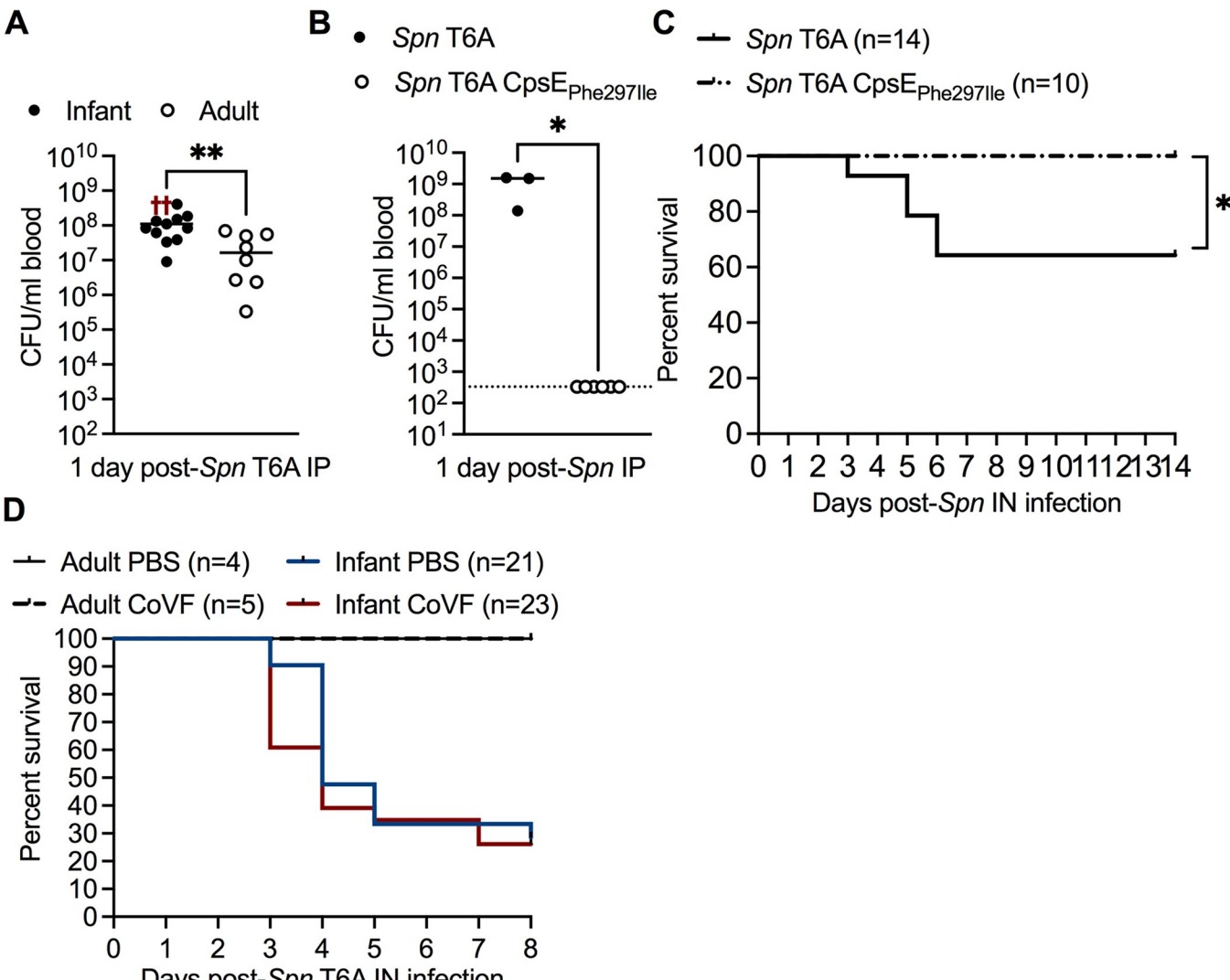

**Fig 2. Increased rate of IPD during early life is independent of systemic control and dependent on *Spn* translocation across the URT barrier. A,** Blood CFU from infant and adult mice intraperitoneally (IP) infected with $10^2$ CFU of *Spn* T6A at 1 dpi (n = 8–13). Data collected from two independent experiments. Red crosses indicate non-surviving mice and were excluded from the statistical analysis. **B,** Blood CFU from infant mice IP infected with *Spn* T6A or *Spn* T6A CpsE$_{Phe297Ile}$ at 1 dpi (n = 3–6). Data collected from one experiment. Statistical significance determined using Mann-Whitney test. *, $p \leq 0.05$; **, $p \leq 0.01$. Data represent individual mice with the median. Dotted line represents limit of detection. **C,** Percent survival (n = 10–14) of infant mice at 14 days post-IN infection with *Spn* T6A or *Spn* T6A CpsE$_{Phe297Ile}$. Data collected from 1–2 experiment(s). Statistical significance determined using Log-rank Mantel-Cox test. *, $p \leq 0.05$. **D,** Percent survival of infant and adult mice treated IP with dPBS or cobra venom factor (CoVF), and IN infected with *Spn* T6A (n = 4–23). Data for adult mice collected from one experiment and infant data are from three independent experiments.

Survival was then used as an indirect measurement of bacterial translocation to the blood. Treatment of infants or adults with CoVF did not significantly alter either the survival rate or onset of septic death compared to control-treated mice (**Fig 2D**). This showed that despite a lower threshold for bacteria required to cause sepsis, adult mice remained resistant to pneumococcal sepsis. Therefore, in contrast to infant mice, *Spn* is impaired in its ability to disseminate across the URT barrier in adults. These results suggested that the age-dependent factor facilitating protection against pneumococcal sepsis during nasal colonization involves restriction of bacterial translocation by the mucosal barrier.

Next, we determined whether bacterial breaching of the URT barrier during early life was dependent on the *Spn* strain. Using survival as a proxy for bacterial spread, we infected infant and adult mice IN with 1 x 10$^3$ or 1 x 10$^6$ CFU, respectively, of *Spn* type 4 (T4), a strain previously associated with invasive pneumococcal disease in humans prior to PCV introduction [19]. In contrast to *Spn* T6A, we observed a 100% survival rate in both infant and adult mice after IN infection with *Spn* T4 (**Fig 3A**). Nasal colonization levels of *Spn* T4 in infant mice were similar to *Spn* T6A infected infants at 3 dpi (**S3A Fig**). These results suggested that either *Spn* T4 was unable to disseminate to the blood in infant mice, or there were strain-dependent

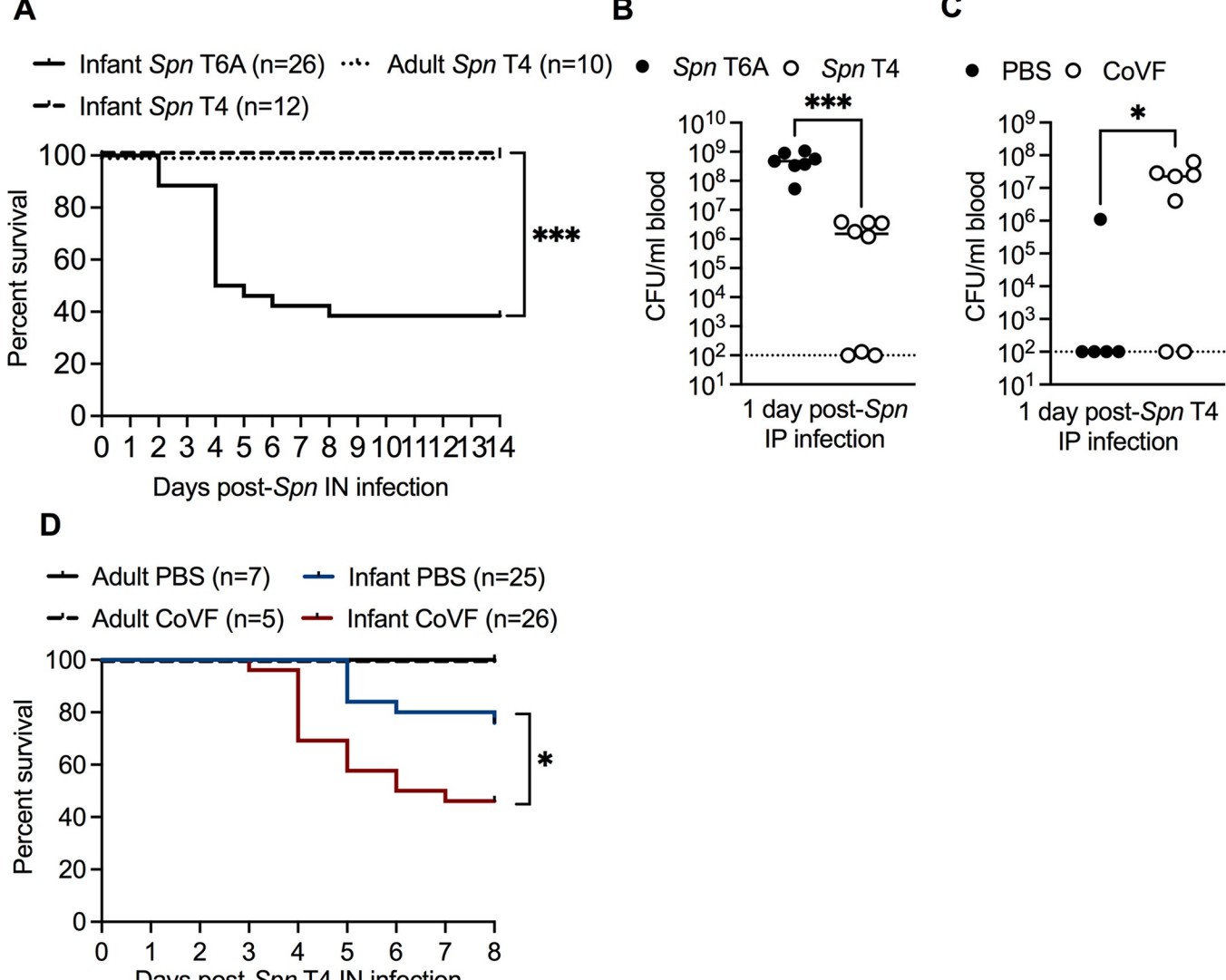

**Fig 3. Factors contributing to invasive infection during early life. A,** Percent survival of infant mice infected IN with *Spn* T6A or infant and adult mice infected with *Spn* type 4 (*Spn* T4) (n = 10–26). Data for *Spn* T6A are re-plotted from **Fig 1A**. Infant *Spn* T4 data collected from two independent experiments and adult *Spn* T4 data collected from one experiment. Statistical significance determined using Log-rank Mantel-Cox test. ***, $p \leq 0.001$. **B,** Blood CFU from infant mice IP infected with *Spn* T6A or *Spn* T4 at 1 dpi (n = 7–8). **C,** Blood CFU from infant mice treated IP with dPBS or CoVF 24 hours prior to IP infection with *Spn* T4 at 1 dpi (n = 5–7). Data are from one experiment. Statistical significance determined using Mann-Whitney test. *, $p \leq 0.05$; ***, $p \leq 0.001$. Data represent individual mice with the median. Dotted line represents limit of detection. **D,** Percent survival of infant and adult mice treated IP with dPBS or CoVF and IN infected with *Spn* T4. Infant data are collected from three independent experiments and adult data are collected from one experiment. Statistical significance determined using Log-rank Mantel-Cox test. *, $p \leq 0.05$.

differences in blood survival. To test whether systemic control of *Spn* T4 is improved compared to *Spn* T6A, we IP infected infant mice with 1 x $10^2$ CFU of either *Spn* T6A or T4, and assessed bacterial density in the blood 1 dpi. Bacterial density in the blood of infant mice IP infected with *Spn* T4 was significantly decreased compared to *Spn* T6A IP infected infants (**Fig 3B**). CoVF treatment prior to IP inoculation of *Spn* T4 in infant mice significantly increased bacterial density in the blood (**Fig 3C**) compared to dPBS-treated controls. These results suggested that the variation in survival rates of infant mice following IN infection with *Spn* T4 versus T6A is likely due to enhanced C3-mediated phagocytosis of *Spn* T4 in the bloodstream, rather than strain-dependent differences in bacterial translocation from the URT. Therefore, we hypothesized that depletion of circulating C3 in *Spn* T4 colonized infant mice, but not adults, would significantly increase rates of septicemia. To test this, we treated infant and adult mice IP with either CoVF or PBS control prior to, and after, IN *Spn* T4 infection and assessed survival 8 dpi. Similar to our observations with *Spn* T6A, CoVF treatment did not impact survival rates of *Spn* T4 colonized adult mice compared to dPBS controls (**Fig 3D**). In contrast, depletion of circulating C3 with CoVF treatment significantly reduced the survival rate of *Spn* T4 colonized infant mice compared to dPBS treated controls (**Fig 3D**). All together, these results demonstrate that the age-dependent limiting factor driving susceptibility to invasive pneumococcal infection in infant mice involves disruption of URT barrier function, which permit the translocation of *Spn* across the epithelium regardless of *Spn* strain. Further, our results demonstrate that *Spn* sepsis during nasal colonization differs among isolates, but is primarily limited by factors that allow survival once in the bloodstream, rather than passage across the epithelial barrier. These factors could include, for example, capsule type and other characteristics involved in evading complement activity.

## Neutrophil-mediated inflammation does not correlate with *Spn* dissemination from the URT

Although *Spn* colonization is typically asymptomatic, the local immune response is characterized by an early influx of neutrophils that is followed by bacteria-clearing monocytes and macrophages [20]. We hypothesized that neutrophil-mediated damage to the respiratory epithelial barrier promoted dissemination of *Spn* from the URT during early life. To test this, we determined URT transcript levels of the neutrophil chemokines, C-X-C motif chemokine ligand 1 (*Cxcl1*) and macrophage inflammatory protein 2-alpha (*Cxcl2*), from mock infected or *Spn* T6A-infected infant and adult mice 2–3 dpi. Infection with *Spn* significantly increased expression of *Cxcl1* (**Fig 4A**), *Cxcl2* (**Fig 4B**) and lipocalin-2 (*Lcn2*) (**Fig 4C**), a marker of inflammation, in infant mice compared to adults. Increased expression of *Cxcl1* and *Cxcl2* was not dependent on whether the infant mice were septic, as non-septic infant mice alone displayed increased expression of *Cxcl1* ($p = 0.0003$) and *Cxcl2* ($p = 0.0007$) compared to adults (**Fig 4A and 4B**). Consistent with elevated expression of neutrophil chemokines, we also found that the proportion of neutrophils in the CD11b$^+$ population (**Fig 4D**), but not monocytes (**Fig 4E**), was increased in nasal tissue of *Spn*-infected infant mice compared to adults. To test whether neutrophils facilitated *Spn* dissemination from the URT, we depleted neutrophils in the URT by treating infant mice with an anti-Gr-1 monoclonal antibody (αGr-1) or isotype (IgG) control, and assessed bacterial spread to the blood and spleen 3 days post-*Spn* IN infection. Treatment with αGr-1 led to a significant reduction in neutrophil numbers in both the blood (**S4A Fig**) and nasal tissue (**S4B Fig**). Although blood monocyte numbers were significantly reduced compared to isotype controls (**S4C Fig**), URT numbers were unaltered (**S4D Fig**). Depletion of neutrophils with αGr-1 treatment in infant mice did not impact nasal colonization (**Fig 4F**) or *Spn* dissemination to the blood (**Fig 4G**) and spleen (**Fig 4H**) compared to isotype treated

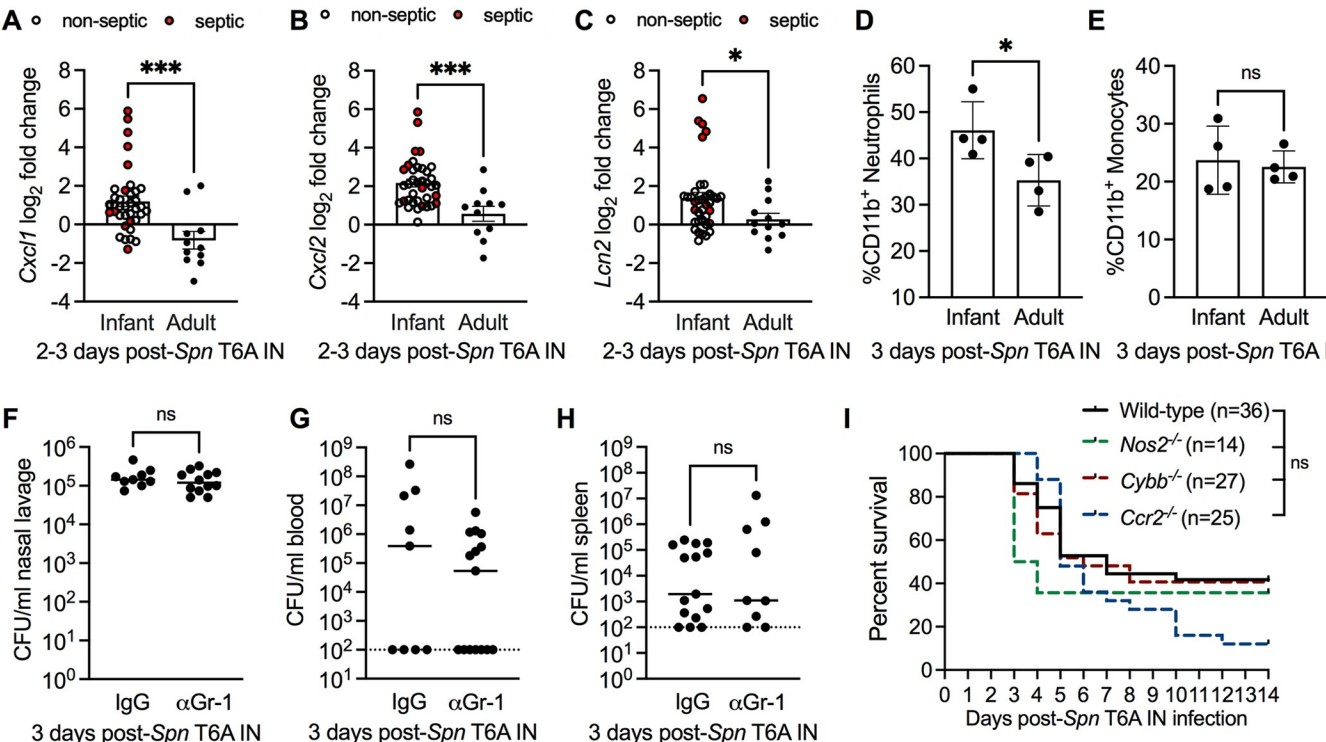

**Fig 4. Neutrophil-mediated inflammation does not correlate with *Spn* dissemination. A,** Transcript levels of *Cxcl1*, (**B**) *Cxcl2* and (**C**) *Lcn2* from IN mock- or *Spn* T6A-infected infant and adult mice at 2–3 dpi (n = 7–37). Data collected from 2–3 independent experiments. Statistical significance determined using unpaired Students *t* test. *, $p \leq 0.05$; ***, $p \leq 0.001$. **D**, Percentage of neutrophils (Live CD11b⁺ Ly6G⁺) and (**E**) monocytes (Live CD11b⁺ Ly6G⁻ Ly6C⁺) in nasal tissue from *Spn* T6A IN infected infant and adult mice (n = 4). Data collected from one experiment. Statistical significance determined using unpaired Students *t* test. *, $p \leq 0.05$; ns, not significant. **F**, Nasal lavage, (**G**) blood and (**H**) spleen CFU from infant mice IP treated with IgG isotype control (IgG) or anti-Gr-1 (αGr-1) antibody and IN infected with *Spn* T6A at 3 dpi (n = 9–12). Data represent individual mice with the median and are collected from 2–3 independent experiments. Dotted line represents limit of detection. Statistical significance determined using unpaired Students *t* test or Mann-Whitney test. ns, not significant. **I**, Percent survival of infant Wild-type (n = 36), *Nos2*⁻/⁻(n = 14), *Cybb*⁻/⁻ (n = 27) and *Ccr2*⁻/⁻ (n = 25) mice infected IN with *Spn* T6A at 14 dpi. Wild-type mice data from 5 independent experiments, *Nos2*⁻/⁻ from 2 independent experiments, *Cybb*⁻/⁻ from 3 independent experiments and *Ccr2*⁻/⁻ from 3 independent experiments. Statistical significance determined using Log-rank Mantel-Cox test. ns, not significant.

controls. However, depletion of neutrophils in the URT was incomplete, which raised the possibility that the remaining neutrophils might be sufficient to cause enough barrier damage to facilitate bacterial spread. Two major effector molecules produced by activated neutrophils in response to bacterial infection are reactive oxygen and nitrogen species (ROS/RNS). Previous work has demonstrated that release of phagocyte-derived ROS/RNS can reduce epithelial barrier integrity [21]. To test whether neutrophil–mediated production of ROS/RNS impacts susceptibility to pneumococcal dissemination in infant mice, we IN infected wild-type, *Cybb*-deficient (lack phagocyte superoxide production) and *Nos2*-deficient infant mice with 10³ CFU of *Spn* T6A and assessed survival 14 dpi. Loss of phagocyte derived superoxide (*Cybb*⁻/⁻) or nitric oxide (*Nos2*⁻/⁻) did not impact survival rates of infant mice compared to wild-type controls (**Fig 4I**). Further, to rule out a role of inflammatory monocytes, we colonized wild-type and CCR2-deficient (*Ccr2*⁻/⁻; defective for monocyte recruitment) infant mice with *Spn* and assessed survival 14 dpi. Loss of monocyte recruitment also did not impact survival rates compared to wild type controls (**Fig 4I**). Together, these results demonstrated that infiltration of inflammatory myeloid cells in response to URT *Spn* colonization does not contribute to rates of septic death during early life.

## Susceptibility to invasive pneumococcal infection during early life is independent of inflammatory stimuli

In addition to recruitment of neutrophils, the pore-forming activity of the sole *Spn* toxin, pneumolysin, has been shown to contribute to barrier damage by triggering necroptotic epithelial cell death in adult mice [22–24]. Since blockade of neutrophil URT influx did not mitigate susceptibility to pneumococcal bacteremia during early life, we hypothesized that pneumolysin-mediated damage to the respiratory epithelium via induction of epithelial cell death facilitated pneumococcal translocation across the barrier. To test this, we IN infected infant mice with $10^3$ CFU of either a pneumolysin-deficient (Δ*ply*) or an isogenic corrected control (*ply*$^+$) type 6A strain. Bacterial burden in the nasal lavage, blood, and spleen was determined at 3 dpi and survival was assessed at 14 dpi. Loss of pneumolysin did not impact bacterial density in the nasal lavage (**Fig 5A**), blood (**Fig 5B**), or spleen (**Fig 5C**) compared to pneumolysin-sufficient infected infants. In accordance with systemic bacterial loads, we observed similar rates of septic death in infants colonized with either Δ*ply* or *ply*$^+$ (**Fig 5D**). These results indicated that pneumolysin-mediated damage to the mucosal barrier does not impact bacterial dissemination from the URT.

Initiation of the inflammatory response and potential barrier damage during *Spn* nasal colonization also involves the activation of Toll-like receptor 2 (TLR-2) signaling following

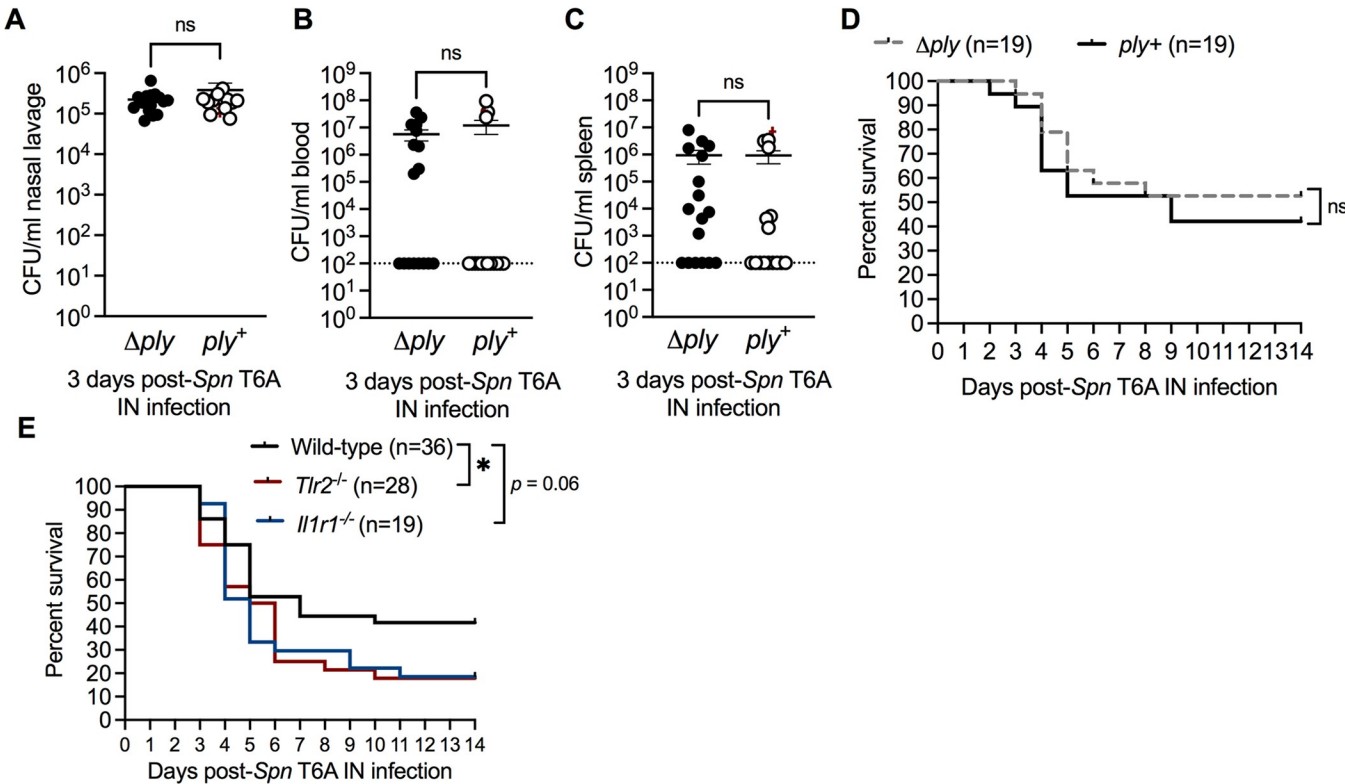

**Fig 5. Susceptibility to IPD is independent of pneumolysin. A,** Nasal lavage, (**B**) blood and (**C**) spleen CFU from infant mice IN infected with *Spn* T6A pneumolysin-deficient (Δ*ply*) or corrected (*ply*$^+$) strain at 3 dpi (n = 16–17). Data represent individual mice with the mean ±SEM and are collected from 3 independent experiments. Statistical significance determined using unpaired Students *t* test or Mann-Whitney test. ns, not significant. **D,** Percent survival of infant mice IN infected with *Spn* T6A Δ*ply* or *ply*$^+$ at 14 dpi (n = 19). Data collected from 3 independent experiments. Statistical significance determined using Log-rank Mantel-Cox test. ns, not significant. **E,** Percent survival of infant Wild-type (n = 36), *Il1r1*$^{-/-}$ (n = 19) and *Tlr2*$^{-/-}$ (n = 28) mice IN infected with *Spn* T6A at 14 dpi. Survival data from Wild-type infant mice are re-plotted from **Fig 4I**. *Il1r1*$^{-/-}$ data from 4 independent experiments and *Tlr2*$^{-/-}$ data are from 3 independent experiments. Dotted line represents limit of detection. Statistical significance determined using Log-rank Mantel-Cox test. *, $p \leq 0.05$.

recognition of bacterial lipoproteins [23], and interleukin 1 (IL-1) signaling by pneumolysin [25]. Therefore, we determined whether inflammation mediated by either TLR-2 or IL-1 signaling contributed to bacterial spread from the URT. To test this we colonized wild-type, *Tlr2*[-/-] and *Il1r1*[-/-] infant mice with $10^3$ CFU of *Spn* T6A and assessed survival at 14 dpi. Infant mice colonized with *Spn* remained susceptible to pneumococcal sepsis regardless of whether TLR-2 or IL-1-dependent signaling was intact (**Fig 5E**). Together, these results suggested that the canonical inflammatory pathways activated during early colonization by *Spn* are not responsible for the increased susceptibility to invasive pneumococcal infection during early life.

## Population bottleneck limits *Spn* translocation from the URT in infant mice

Next, we characterized the magnitude of the epithelial breach in infants by determining whether bacterial translocation to the bloodstream was characterized by a tight or wide population bottleneck. To determine the proportion of the nasopharyngeal *Spn* population that disseminates to the blood, we colonized infant mice with a chromosomally barcoded library of *Spn* T6A (**S5A Fig**) and determined clonal diversity in blood relative to that in nasal lavages at 3 dpi. While approximately 12% of nasopharyngeal clones disseminated to the blood (**Fig 6A**), the frequency of the most abundant clone in the blood was >95% in the majority of mice (**Fig 6B**), indicating the dominance of only a single clone. The success of a certain clonal lineage in the blood or nasopharynx was not dependent on its abundance in the other anatomical niche (**Fig 6C and 6D**). Additionally, the success of a single clone was independent of any inherent fitness difference in the inoculum, as infant mice infected IP with the *Spn* T6A library exhibited >1000 unique clones in the blood with the most abundant clone present at a frequency of only 1% (**S5B–S5C Fig**).

To further characterize the bottleneck, infant mice were treated IP with CoVF to determine whether complement-mediated clearance of bacteria in the blood affected clonal diversity. Treatment with CoVF had no impact on nasal *Spn* colonization (**S5D Fig**), bacterial dissemination to the blood (**S5E Fig**) or nasopharyngeal clonal diversity indicated by the Hills $N_1$ coefficient (**S5F Fig**) compared to PBS controls. CoVF-treated infant mice also exhibited a significant decrease in clonal diversity in the blood relative to nasal lavage (**Fig 6A**). Additionally, the dominance of a single clone at >95% in the blood remained despite depletion of circulating complement (**Fig 6B**). Similar to PBS treated controls, the clonal success of a single lineage in the blood was not a function of its abundance in the nasal lavage (**Fig 6C**). The lack of an effect of CoVF treatment implied that the population bottleneck occurred prior to *Spn* accessing the bloodstream. Together, our data suggested dissemination of *Spn* was a rare event and did not involve substantial damage to the nasal epithelium.

## Age-associated disruption in barrier function

To further explore age-dependent differences in barrier function, we utilized our published RNA-sequencing screen to perform additional pathway analysis on URT samples collected from mock-infected infant and adult mice [6]. KEGG pathway analysis of significantly downregulated genes in infants revealed enrichment in the *Tight junction* pathway (**Fig 7A**). Using quantitative real time-PCR (qRT-PCR) we confirmed that expression of the actin related protein 2/3 complex subunit 1b (*Arpc1b*) and claudin 8 (*Cldn8*), two *Tight junction* pathway genes identified from the screen, was significantly reduced in early life (**Fig 7B and 7C**). The dampened expression of genes involved in the *Tight junction* pathway suggested that paracellular permeability of the URT barrier was compromised in infant mice. To directly assess URT

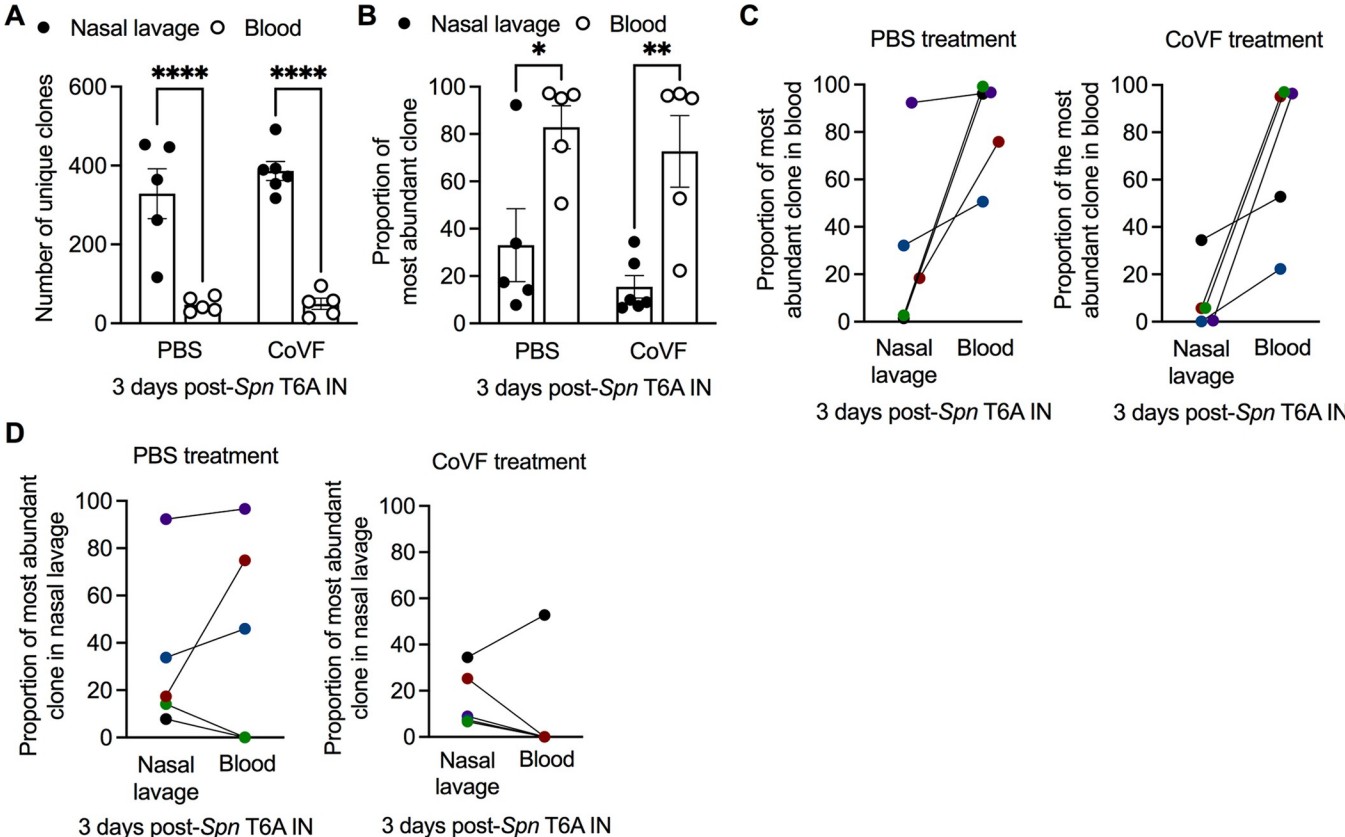

**Fig 6. Population bottleneck limits *Spn* translocation from the URT in infant mice. A-D,** Infant mice IP treated with dPBS or CoVF and IN infected with a *Spn* T6A chromosomally-barcoded clonal library at 3 dpi (n = 5). **A,** Number of unique clones and (**B**) proportion of most abundant clone in the nasal lavage and blood. Data represent individual mice with the mean ±SEM. Statistical significance determined using One-way ANOVA with a Sidak's post-hoc multiple comparisons test. *, $p \leq 0.05$; **, $p \leq 0.01$; ****, $p \leq 0.0001$. **C,** Proportion of the most abundant clone in the blood and its corresponding proportion in the nasal lavage from each individual mouse. **D,** Proportion of the most abundant clone in the nasal lavage and its corresponding proportion in the blood from each individual mouse. Data are collected from two independent experiments.

barrier integrity, we IN inoculated naïve 13-day-old infant mice, an age corresponding to the peak onset of sepsis, and adult mice with 20 mg/kg of 4 kDa fluorescein isothiocyanate (FITC)-dextran or PBS control, and measured plasma FITC fluorescence at 1 hour post-treatment. In accordance with reduced expression of tight junction genes, infant mice had significantly higher FITC fluorescence in plasma compared to adults, indicative of increased paracellular permeability (**Fig 7D**).

Considering infant mice displayed a significant reduction in URT barrier integrity, we wondered whether humans also exhibited an age-dependent regulation of tight junction genes during early life. To test this, we assessed expression of the tight junction genes identified in **Fig 7A** in nasopharyngeal samples collected from healthy infants at 11 time-points in the first year of life (2 hours after birth, 24 hours, 7 and 14 days, and 1,2,3,4,6,9, and 12 months of age). Of the 38 murine genes assessed, 37 were identified in the human dataset. When comparing week 1 and year 1 samples, 10 genes exhibited significantly higher expression with age (**Fig 7E**). For example, expression of *ARPCB1* was significantly increased at 1 year of age compared to 1 week (**Fig 7F**). Expression of *ARPCB1* also gradually increased throughout the first year of life, which suggested natural maturation rather than initial impairment (**Fig 7G**). While not all of the tight junction genes in infant humans followed the same transcriptional pattern observed

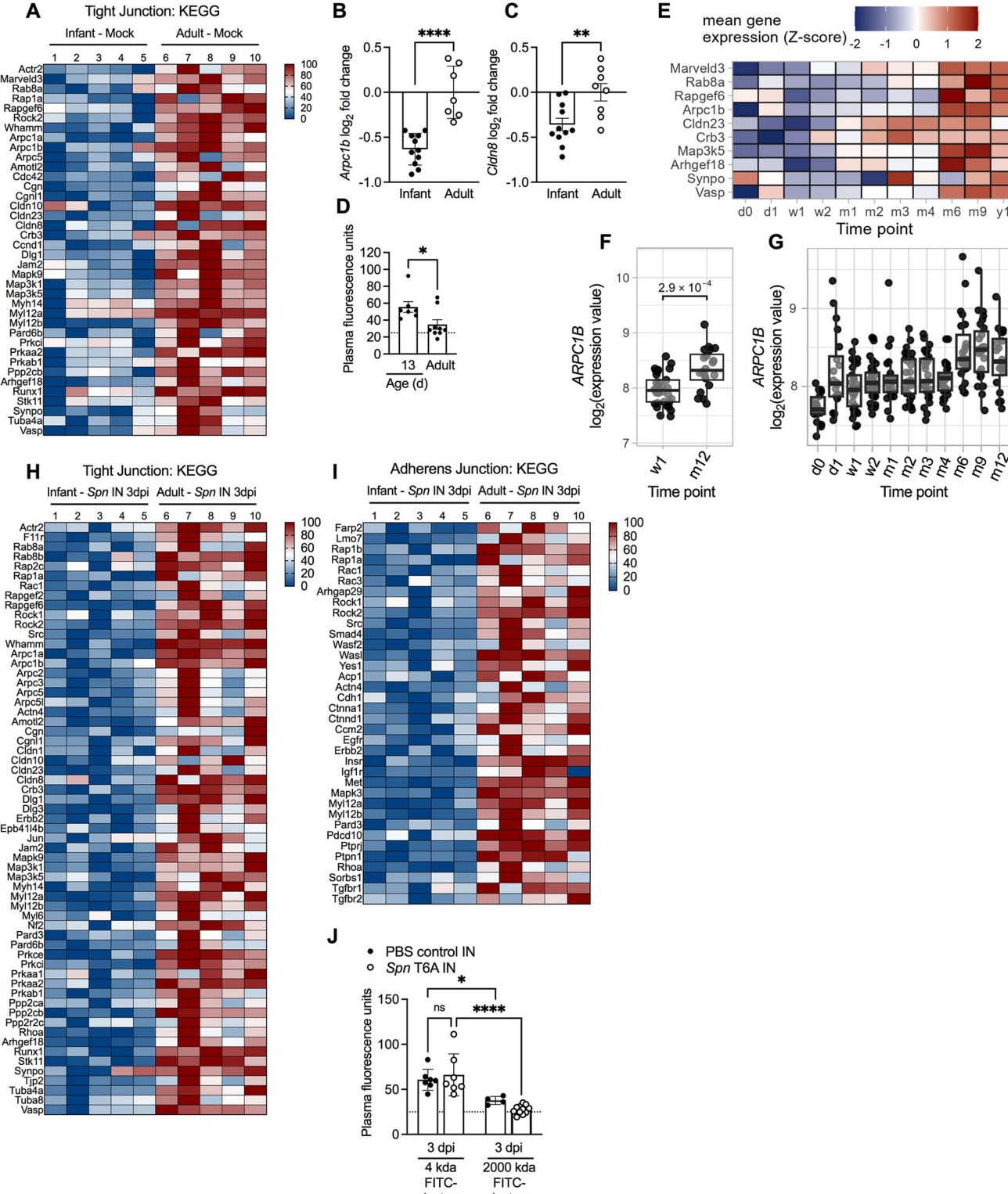

**Fig 7. Age-associated disruption in URT barrier function. A,** Heat-map representing the fold-change in expression of genes involved in the KEGG Tight Junction Pathway from mock treated infant vs. adult mice. Expression from individual mice was normalized to the smallest (0%) and largest value (100%) for each gene and presented as a percentage within this range (n = 5). **B,** Transcript expression of *Arpc1b* and (**C**) *Cldn8* from naïve 7-day-old infant and adult mice (n = 7–11). Data collected from two independent experiments. Statistical significance determined using unpaired Students *t* test. **, $p \leq 0.01$; ****, $p \leq 0.0001$. **D,** Plasma fluorescence from naïve 13-day-old infant and adult mice 1-hour post IN treatment with 20 mg/kg of 4 kDa FITC-

dextran (n = 7–9). Data collected from 2–3 independent experiments. Statistical significance determined using Mann-Whitney test. *, $p \leq 0.05$. Dotted line average fluorescence from PBS treated infant and adult mice. **E,** Heatmap of the Z-transformed mean gene expression values at each time point. The log$_2$-transformed gene expression values were averaged across samples per time point and then Z-scores for these mean expression values were determined. Genes showing significantly higher transcript levels at 12 months (m12) compared to 7 days (w1) were visualized. **F,** Transcript levels of *ARPC1B* in nasopharyngeal samples from healthy infants at 7 days (w1) and 12 months (m12) of age. **G,** *ARPC1B* in nasopharyngeal samples from healthy infants at 11 time-points (2 hours after birth (d0), at 24 hours (d1), at 7 (w1) and 14 days (w2) and at 1, 2, 3, 4, 6, 9, and 12 months (m1, m2, m3, m4, m6, m9, and m12). N = 286 samples. Data represent log$_2$-transformed intensity values generated using microarray. Statistical significance was determined using a linear mixed effects model to adjust for repeated measures (*lmerTest* R-package) and included time point (w1 or m12) as fixed effect and subject as a random intercept. P-values were adjusted for multiple testing using the Benjamini-Hochberg-method. **H,** Heat-map representing the fold-change in expression of genes involved in the KEGG Tight junction and (**I**) Adherens junction pathways from *Spn* type 23F IN infected infant and adult mice. Expression from individual mice was normalized to the smallest (0%) and largest value (100%) for each gene and presented as a percentage within this range (n = 5). **J,** Plasma fluorescence units 1-hour post-IN treatment with 20 mg/kg of 4 kDA or 2000 kDa FITC-dextran from mock or *Spn* T6A IN infected infant mice (n = 4–11). Data collected from one experiment. Statistical significance determined using One-way ANOVA with a Sidak's post-hoc multiple comparisons test. *, $p \leq 0.05$; ****, $p \leq 0.0001$; ns, not significant. Dotted line average fluorescence from PBS treated infant and adult mice.

in infant mice, the high proportion of genes showing significantly increased expression with age suggested that reduced URT barrier integrity might be a feature of early life development.

Next, we wanted to assess whether URT *Spn* colonization, in the absence of invasive infection, augments epithelial barrier permeability during early life. To test this we performed RNA-sequencing on URT samples collected from infant and adult mice 3 dpi with a non-invasive strain, *Spn* type 23F. KEGG pathway analysis showed that the significantly down-regulated genes in *Spn* infected infant mice compared to infected adults were enriched for pathways involved in barrier integrity and included both *Tight junction* and *Adherens junction* proteins (**Fig 7H–7I**). The expansion of additional down-regulated genes corresponding to the *Tight junction* pathway, combined with genes involved in the *Adherens junction* pathway from *Spn* infected infant mice suggested that *Spn* colonization exacerbates the compromise in barrier permeability during early life. To test this, infant mice were either mock- or *Spn* T6A-infected and treated IN with 20mg/kg of 4kDa FITC-dextran, 2000kDa FITC-dextran, or PBS control at 3 dpi. Regardless of the size of dextran administered, we did not detect any difference in the level of plasma FITC fluorescence from *Spn*-infected infant mice compared to mock-infected controls (**Fig 7J**). Further, in contrast to treatment with the 4kDa dextran, the level of FITC fluorescence in infant mice treated with the 2000kDa dextran was below the limit of detection, which indicated that the egress of large molecules was rare (**Fig 7J**). Together, these results suggested that reduced URT barrier function during early life is a consequence of postnatal development and independent of *Spn* colonization.

## Decreased *Spn* association with the URT epithelium with age

Our previous results suggested that reduced URT barrier function follows an age-dependent pattern. To determine the extent of decreased barrier integrity during the first 3 weeks of life, we assessed barrier permeability by IN treatment with 20 mg/kg of 4kDa FITC-dextran in naïve infant mice at 7, 10, 12, 14, 16 and 21 days of age, and compared plasma fluorescence to our previously collected adult data (**Fig 7D**). While plasma FITC levels were generally increased throughout the first two weeks of life, a statistically significant difference was observed only in 14-day-old mice compared to adults (**Fig 8A**). To determine whether this correlated with susceptibility to septic infection, we IN infected 15-16-day-old mice with *Spn* T6A and assessed survival. While colonization of 10 day-old infant mice led to a survival rate of 38% at 14 dpi (**Fig 1A**), colonizing infants 5–6 days later increased the survival rate to 94% (**Fig 8B**).

Assessment of FITC fluorescence in nasal lavages collected from 7-, 10-, 12- and 14-day-old mice revealed significantly elevated levels compared to adults, which suggested delayed clearance of FITC-dextran from the nasopharynx during early life (**Fig 8C**). However, by 16 days of

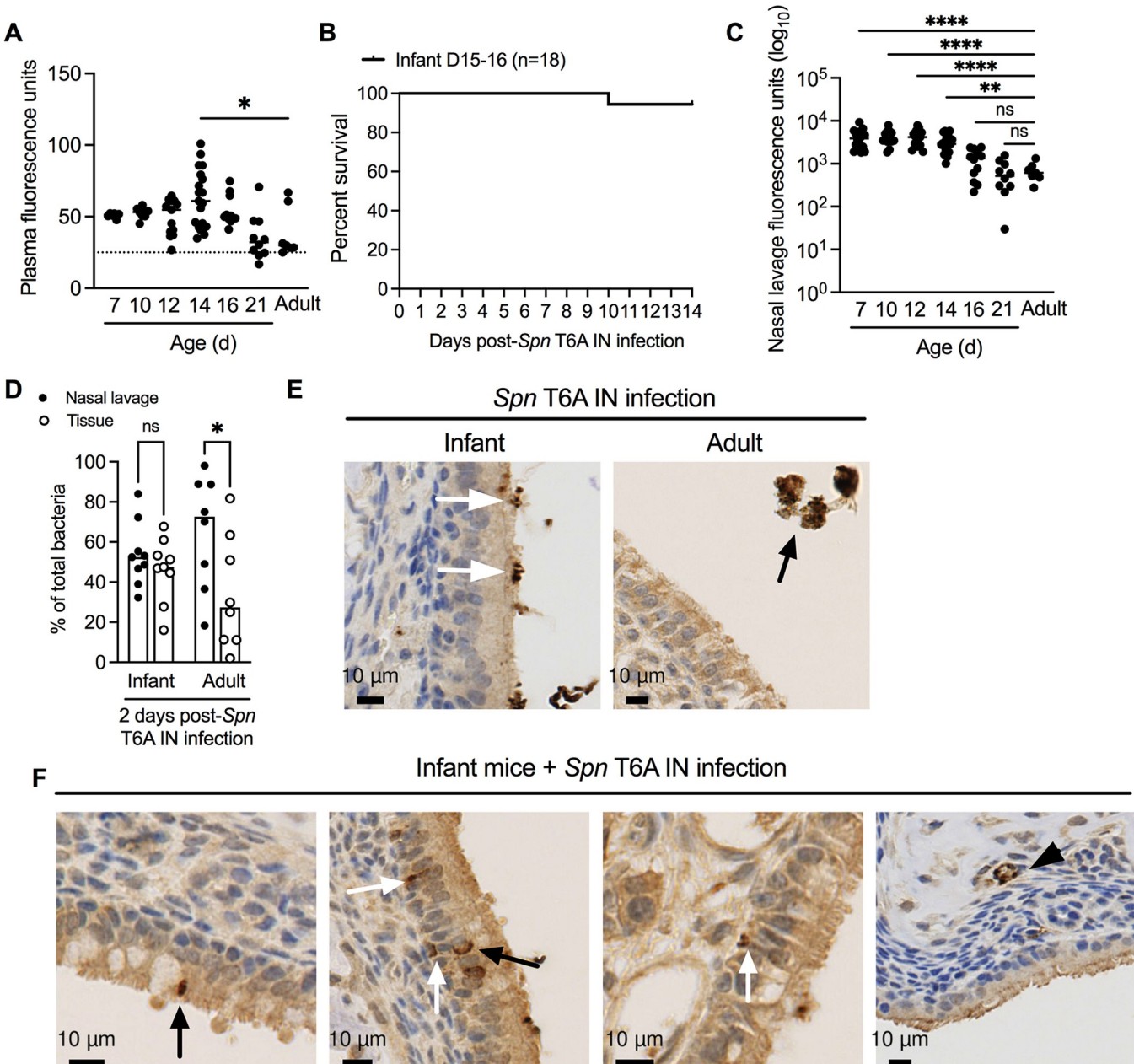

**Fig 8. Decreased *Spn* proximity to the nasal epithelium with age. A,** Plasma fluorescence units from naïve 7 (plasma pooled n = 2–3), 10, 12, 14, 16, 21-day old infant and adult mice 1-hour post-IN treatment with 20 mg/kg of 4 kDA FITC-dextran (n = 6–19). Adult plasma data re-plotted from **Fig 7D**. Data represent individual mice with the median and are collected from two independent experiments. Statistical significance determined using One-way ANOVA with a Dunn's post-hoc multiple comparisons test. *, $p \leq 0.05$. Dotted line average fluorescence from PBS treated infant and adult mice. **B,** Percent survival of 15-16-day-old infant mice IN infected with *Spn* T6A at 14 dpi (n = 18). Data collected from two independent experiments. **C,** Nasal lavage fluorescence units from mice in **Fig 8A**. Statistical significance determined using One-way ANOVA with a Dunn's post-hoc multiple comparisons test. **, $p \leq 0.01$; ****, $p \leq 0.0001$; ns, not significant. **D,** Percentage of nasal lavage and tissue associated *Spn* T6A from infant and adult mice at 2 dpi (n = 8–9). Data represent individual mice with the median and are collected from two independent experiments. Statistical significance determined using One-way ANOVA with a Sidak's post-hoc multiple comparisons test. *, $p \leq 0.05$; ns, not significant. **E-F,** Representative nasal tissue sections from *Spn* T6A colonized infant or adult mice stained for *Spn* T6A capsule (dark brown) and counterstained with hematoxylin at 3 dpi. **E,** *Spn* T6A capsule stain lining the respiratory epithelium of infant mice (white arrows) and in the lumen (black arrow) of adult mice. **F,** Paracellular (black arrows), sub-mucosa (white arrows) and blood vessel (black arrowhead) staining of *Spn* T6A capsule from infant mice.

age, FITC fluorescence decreased to levels in adult mice, indicating improved clearance (**Fig 8C**). These results indicated that potential defects in nasal mucociliary clearance during the first two weeks of life might facilitate closer attachment of *Spn* to the epithelium and, subsequently, enable dissemination to the bloodstream.

To assess whether age impacts URT localization of *Spn*, we IN infected 10-day-old infants and adult mice with *Spn* T6A and collected nasal lavages and tissue at 2 dpi, a time-point prior to the peak onset of sepsis in infants, for bacterial enumeration. While the percentage of *Spn* in the nasal lavage and tissue was equal in infant mice, adult mice had significantly more *Spn* in the nasal lavage than tissue (**Fig 8D**). Complementary to these findings, serotype-specific staining of *Spn* in nasal tissue sections from infected infant and adult mice showed intimate attachment of bacteria with the respiratory epithelium in infants (white arrow); whereas in adults, staining of *Spn* was restricted to the lumen (black arrow) (**Fig 8E**). In bacteremic confirmed infant mice, *Spn* staining within vessels of the sub-mucosa was present (black arrowhead) (**Fig 8F**). However, regardless of whether infant mice were bacteremic, we observed *Spn* staining on the basal side of the epithelium (white arrows) and between goblet and ciliated cells (black arrows), consistent with paracellular bacterial translocation only in infant mice (**Fig 8F**). Collectively, these results suggest that greater epithelial dysfunction in early life, which affects permeability and *Spn* proximity to the respiratory epithelium, enhances the likelihood of successful bacterial dissemination to the bloodstream.

## Discussion

In this study, we demonstrated that age-dependent disruption of URT barrier function underlies susceptibility to invasive *Spn* infection in infant mice. While adult mice successfully restricted *Spn* translocation from the URT, infant mice exhibited increased *Spn* translocation to the bloodstream, regardless of bacterial strain. Increased dissemination of *Spn* in infant mice was independent of barrier damage mediated by either host inflammatory responses or bacterial production of the pore-forming toxin pneumolysin. Rather, infants displayed a significant reduction in expression of tight junction proteins that corresponded with increased URT barrier permeability and delayed mucociliary clearance. A similar transcriptional signature was also demonstrated in nasopharyngeal samples collected from healthy infants during the first year of life, suggesting decreased barrier integrity might be a characteristic feature of postnatal development in the URT. While URT barrier integrity and function increased with age, there was a concurrent decrease in both *Spn* proximity to and within the nasal epithelium, and occurrence of septic death in mice. Together, our data demonstrate a vulnerable period during postnatal development when decreased nasal barrier defenses are exploited by *Spn* to facilitate its migration across the epithelium and cause invasive disease.

Early colonization with *Spn* involves the stimulation of multiple innate inflammatory pathways that become activated by either host recognition of bacterial PAMPs or direct stimulation by release of bacterial virulence factors [26]. Pneumolysin, peptidoglycan, and teichoic acid have been shown to mediate substantial levels of cytotoxicity and inflammation in models of invasive disease [27–30]. While we observed increased expression of URT inflammatory markers in infant mice during early *Spn* infection, blockade of inflammatory cell infiltration, innate signaling pathways, or bacterial virulence factors was inadequate to protect infant mice from developing *Spn* sepsis. These results suggested that the stereotypic inflammatory responses evoked during early *Spn* colonization do not contribute to age-related bacterial spread. However, considering *Spn* colonization involves activation of multiple innate immune pathways, loss of an individual component may be insufficient to significantly dampen the overall inflammatory response. Therefore, we speculated that if bacterial spread relied on robust

barrier breakdown, then the *Spn* population in the blood of septic mice should mirror the clonal diversity of the nasopharyngeal population. In contrast to the nasopharyngeal population, we observed the dominance of a single clone in the blood of septic mice, which indicated that mucosal passage of *Spn* to the bloodstream was restricted by a tight bottleneck. Our observation that 4kDa, but not 2000kDa FITC-dextran accessed the serum from the URT further supports that invasion of large structures, like bacteria, is a rare event. This result was consistent with our previous observation that demonstrated concurrent influenza A virus infection imposed additional constraints on the population bottleneck during *Spn* passage to the bloodstream despite increased URT inflammation [31]. Previous studies investigating *Spn* and *Haemophilus influenzae* pathogenesis also identified single-cell bottlenecks in the establishment of bacteremia [32,33]. Both studies described an eclipse phase that was followed by the expansion of one 'founder' bacterium, implicating a systemic reservoir in driving the onset of bacteremia [32,33]. Recently, splenic macrophages were identified as an *Spn* reservoir in models of bacteremia [34,35]. Therefore, it was possible that a large number of *Spn* migrate across the nasal epithelium to the bloodstream, but only a limited number of clones successfully evade clearance by phagocytes, leading to their expansion in the blood. However, when phagocyte-mediated clearance of *Spn* was inhibited by complement depletion, we observed no discernable effect on clonal diversity in the blood of septic mice. This suggested that the existence of a systemic *Spn* reservoir during natural infection in infant mice was unlikely, and instead indicated that the bottleneck occurred prior to *Spn* accessing the bloodstream.

As a prerequisite to IPD, stable *Spn* colonization of the nasopharynx is achieved through intimate attachment to the epithelium [2]. To mitigate bacterial adherence, the epithelial barrier produces antimicrobial peptides and mucus to limit bacterial growth and entrap bacteria, respectively, facilitating their removal via mucociliary action. The increased localization of *Spn* to the nasal epithelium in infant mice suggested that one or both of these initial defenses might be impacted during early life development. We previously demonstrated that postnatal development is associated with reduced URT production of the most abundant epithelial-derived antimicrobials [6]. Results from this study showed an age-dependent delay in nasal mucociliary clearance. In accordance with reduced mucociliary clearance, previous studies also demonstrated a postnatal delay in tracheal ciliated cell abundance and cilia-generated flow in infant mice [36,37]. Together, these results suggest that decreased antimicrobial proteins and impaired mucus flow in infant mice lead to increased survival and retention of *Spn* in the URT, thereby facilitating tight association with the mucosal barrier. Similarly, resistance to invasive *Escherichia coli* K1 infection in neonatal rats corresponded with increased physical separation of the bacterium from enterocytes of the small intestine, which coincided with an age-dependent maturation of the mucus barrier [38]. Additionally, age-dependent alterations in intestinal mucus layer thickness, combined with reduced production of Paneth cell-derived antimicrobials, was proposed to contribute to increased susceptibility of infant mice to disseminated non-typhoidal *Salmonella* infection [39]. Collectively, these results suggest that global breakdown in mucosal barrier defenses designed to physically restrict bacterial contact with the epithelium during early life promotes susceptibility of infants to enteric and respiratory pathogens.

The decreased expression of multiple URT genes involved in the tight junction pathway from mock- and *Spn*-infected infant mice suggested that age- and/or *Spn*-dependent modulation of barrier integrity facilitate paracellular movement of *Spn*. While we observed a significant increase in barrier permeability in infants compared to adult mice at baseline, barrier permeability between mock- and *Spn*-infected infants remained similar. These results suggested that the alteration of nasal barrier integrity in *Spn*-infected infant mice is primarily a consequence of early life development. This differs from earlier studies, which proposed that

targeted disruption of cell junctions by *Spn* facilitated paracelluar spread [12,40–42]. However, all of these studies were performed using *in vitro* cell culture or adult animal models, where the role of age and development is not considered. In a neonatal mouse model of group B streptococcus (GBS) infection, increased Wnt activity, required to support a rapidly growing epithelium, was proposed to contribute to the alteration of cell-cell junctions in the intestine and choroid plexus epithelia that favored bacterial translocation [43]. We previously showed that the URT of infant mice exhibited a similar transcriptional profile with increased expression of genes involved in the Wnt signaling pathway [6]. Therefore, while the global changes in mucosal barrier integrity during development enable the physical maturation of the expanding airway and intestinal epithelium, they also create a window of vulnerability. This window can be exploited by pathogens capable of breaching initial barrier defenses and evading phagocyte-mediated clearance once in the bloodstream.

While staining of *Spn* between epithelial cells lining the nasal epithelium of infants was consistent with paracellular movement, we cannot exclude a role for transcellular transport of *Spn* across the barrier. Previous studies demonstrated that transmigration of *Spn* across human respiratory epithelial cells is mediated by interaction of its adhesion, choline-binding protein A (CbpA; also known as PspC), with the secretory component on the polymeric immunoglobulin receptor (pIgR) [13,44]. However, the extent of this interaction in facilitating transcytosis across the epithelium in mouse models remains unclear given the species-specific binding of pIgR to CbpA [45]. Another endocytic route of *Spn* transcytosis involves recognition of phosphorylcholine on the surface of *Spn* with the platelet activating factor receptor (PAFr) expressed on the respiratory epithelium [46,47]. Whether postnatal development corresponds with increased expression of these host receptors is an area of active investigation. During early life development, high endocytic activity of immature enterocytes contributes to increased barrier permeability of the intestinal epithelium [48]. This heightened endocytic activity is crucial for facilitating the passive transfer of maternal antibodies, and microbial and dietary antigens, which drive the maturation and education of the developing immune system [48]. Similar to the gastrointestinal tract, the increased barrier permeability observed in the URT during early life in this study may also contribute to immune development.

For decades, the encapsulated bacteria *Streptococcus pneumoniae*, *Haemophilus influenzae*, and *Neisseria meningitidis* have been the predominant bacterial pathogens responsible for a large proportion of childhood mortality worldwide [49]. Yet, asymptomatic colonization by these respiratory, opportunistic pathogens in young children is common. The age-dependent factors that enable the abrupt transition from a commensal lifestyle to a pathogenic one in children are unknown. The results from this study demonstrated that age-dependent disruptions in URT mucosal barrier defenses, rather than systemic phagocyte control, are responsible for predisposing young children to invasive infection. In contrast to the gastrointestinal tract, few studies have investigated the consequences of postnatal development on airway epithelium function and susceptibility to pathogens. This study highlights the need for a better understanding of the age-dependent mechanisms that contribute to respiratory barrier dysfunction, which can be exploited by pathogens to facilitate their dissemination and cause disease.

## Materials and methods

### Ethics statement

All animal experiments followed the guidelines provided by the National Science Foundation Animal Welfare Act (AWA) and the Public Health Service Policy on the Humane Care and Use of Laboratory Animals. The Institutional Animal Care and Use Committee (IACUC) at

New York University School of Medicine oversees the welfare, well-being and proper care and use of all animals, and approved the protocol used in this study, IA16-00538.

## Bacterial strains

For mouse infections, all pneumococcal strains were grown in tryptic soy (TS) broth (BD) at 37°C without aeration to an optical density of 1.0 at 620 nm. For *in vivo* enumeration of bacteria, pneumococci were incubated on TS plates supplemented with 100 μl of catalase (30,000 U/ml; Worthington Biomedical) and 200 μg/ml of streptomycin or spectinomycin (Sigma) at 37°C in 5% $CO_2$ overnight. Streptomycin resistant *Streptococcus pneumoniae* type 4 (P2406) and type 6A (P2431), used in this study, were made previously [50]. For the RNA-Sequencing screen, infant and adult mice were infected with a streptomycin resistant type 23F strain (P1397) described previously [25].

An inframe, unmarked deletion of *ply* was constructed by generating a PCR product on genomic DNA from strain P2408 (*ply*::Janus) and P1726 (*ply*-deficient) using primers 1,000 bp upstream (5′-CGCCCTTGCTCTGGTTAAAAAAAGA-3′) and downstream (5′-ATCTGGA TCACCTTTTTTAGCTGC-3) of Ply. To construct the corrected mutant (*ply⁺*), we obtained a PCR product using genomic DNA from *Spn* T6A (P2431) as a template. The pore-forming phenotype of each construct was confirmed with a horse erythrocyte lysis assay [51].

A bacterial colony (*Spn* T6A; strain P385) was mouse passaged at $1 \times 10^5$ CFU via the IP route in order to apply selective pressure for survival in the blood. Colonies obtained from blood cultures of these mice were saved and mouse passaged again via the IP route at a dose of $1 \times 10^2$ CFU. After two mouse passages, we obtained from the bloodstream P2805 with a single amino acid substitution in CpsE. We infected infant mice IN with $1 \times 10^3$ CFU of either P385 or P2805 to assess the contribution of capsule amount in promoting pneumococcal bacteremia following URT colonization.

## Construction of *Spn* T6A barcoded library

The chromosomally barcoded *Spn* 6A library was constructed as before [52]. Briefly, 7-nt barcodes of the sequence NNMCAATGNNMCAAN were obtained from a pE539 plasmid vector library. This plasmid library contains homology to the *Spn* IgA1 protease gene *iga* and a spectinomycin resistance cassette. This plasmid library was a collection of 3,725 uniquely barcoded cells. The uniquely-barcoded pooled plasmid library was transformed into *Spn* T6A (P376) using homologous recombination and the *Spn* transformants were selected on TS plates supplemented with 100 μl of catalase (30,000 U/ml) and 200 μg/ml spectinomycin. A total of 3060 unique barcoded clones were obtained and the resulting barcoded *Spn* library was grown, sequenced and stocked at -80°C.

## *Spn* Library sequencing

The sequencing of barcoded *Spn* was performed as previously described [52]. Genomic DNA from samples was isolated using MasterPure Complete DNA & RNA Purification Kit (Lucigen, Middleton, WI) as per manufacturer's instructions. Barcodes were amplified from genomic DNA using Nested PCR; wherein, the first step consisted of amplifying the *iga* region (5 cycles) followed by amplification of the barcodes (35 cycles). Primers used for amplification of the barcodes contained the adapters to be used for sequencing library preparation. These amplicons were then purified using QIAquick PCR purification kit (Qiagen, Germantown, MD) as per manufacturer's instructions. Purified samples were then shipped to Azenta Life Sciences (South Plainfield, NJ) for sequencing using their Next-Gen Amplicon-EZ service.

## Analysis of sequencing data

The sequencing data was analyzed as previously published [52]. Reads were aligned to a reference sequence using Python. First, trimmomatic was used for quality control to trim adaptor sequences and low-quality bases from the reads (sliding window size: 3, sliding window quality: 20, leading and trailing quality: 15, minimum length: 75). The reads were then aligned to a reference sequence by BWA (Matching Score: 10, Mismatch Penalty: 2) and outputted in a. sam file. The remainder of the analysis was done using R [52]. The barcode sequence was extracted from the aligned reads by concatenating bases at known variable positions while filtering out incomplete or ambiguous barcodes. A table detailing each barcode detected and the number of times it was found was compiled. To account for variability in the number of total reads, we standardized samples by computing rarefaction and extrapolation of clonal diversity using iNEXT. The clonal diversity was expressed using Hill numbers with q = 0 ('clonal richness' or number of unique clones present) and q = 1 (Hill's $N_1$). Clonal diversity index (H) was calculated as $H = -\Sigma pi.ln(pi)$ where $p_i$ denotes the proportion of the population made up of the clone $i$.

## Mouse strains

Male and female C57BL/6J (Jax stock #000664), C57BL/6J $Nos2^{-/-}$ (B6.129P2-$Nos2^{tm1Lau}$/J; Jax stock #002609), C57BL/6J $Cybb^{-/-}$ (Jax stock #002365; B6.129S-$Cybb^{tm1Din}$/J), C57BL/6J $Ccr2^{-/-}$ (Jax stock #004999; B6.129S4-$Ccr2^{tm1Ifc}$/J), C57BL/6J $Il1r1^{-/-}$ (Jax stock #003245; B6.129S7-$Il1r1^{tm1Imx}$/J) and C57BL/6J $Tlr2^{-/-}$ (Jax stock #004650; B6.129-$Tlr2^{tm1Kir}$/J) mice were purchased from Jackson Laboratory (Bar Harbor, Maine) and each colony was bred and maintained in a conventional animal facility. Pups were housed with the dam until 3-weeks of age. The Institutional Animal Care and Use Committee of New York University Medical Center approved all animal experiments.

## Mouse infections

Infant (7-, 10- and 15-16-day-old) and adult mice were infected with either $1x10^3$ or $1x10^5$ CFU, respectively, of $Spn$ in 3 μl or 10 μl of sterile phosphate-buffered saline (PBS), respectively, by intranasal instillation. Adult mice infected with $Spn$ T4 were inoculated with $1x10^6$ CFU in 10 μl of PBS. The inoculum dose was calculated for 10x the 50% colonizing dose for infant and adult mice. For intraperitoneal (IP) infections infant and adult mice with infected with $1x10^2$ CFU of $Spn$ in either 40 μl or 100 μl of sterile Dulbecco's Phosphate Buffered Saline (dPBS), respectively. For $Spn$ T6A library experiments, infant mice were IN inoculated with $1x10^5$ CFU or IP inoculated with $1x10^4$ CFU. All mice were euthanized by $CO_2$ asphyxiation followed by cardiac puncture to collect blood. To assess URT bacterial load, the trachea was cannulated and lavaged with 200–400 μl of sterile PBS. The spleen was removed and collected in 1 mL of sterile PBS with (3–4) 2.7 mm silica beads and homogenized in a bead beater for 45 seconds. For quantification of nasal lavage and tissue bacterial density, the tracheas of $Spn$-infected infant and adult mice were lavaged with 1 mL of sterile PBS. To isolate nasal tissue, heads were detached and the skin, lower jaw, tongue, eyes, and incisors were removed. The skull was cut laterally, starting behind the ears, and the posterior skull and brain tissues were removed. The remaining tissue was cut into small pieces and smashed through a 100 μm cell strainer and washed with either 1 mL (infant mice) or 2 mL (adult mice) of sterile PBS. Blood, spleen, nasal lavage and nasal tissue samples were subsequently serially diluted 10-fold and 10 μl, droplets were plated in triplicate on TS plates supplemented with 100 μl of catalase (30,000 U/ml; Worthington Biomedical) and appropriate antibiotics. For nasal tissue 100 μl was also spread onto TS plates supplemented with catalase and appropriate antibiotics.

## Immunoblot for 6A capsule

For the preparation of immunoblot samples, P385 and P2805 were grown in tryptic soy broth at 37˚C to an optical density of 1.0 at 620 nm. 1 mL of bacterial culture was centrifuged for pellet and re-suspended in 200 μL of 1% Triton-X 100 in 1X PBS for cell lysis. 2 μL of proteinase K (50 μg/μL) and 28 μL of 1X PBS were added to 180 μL of the lysate and incubated at 65˚C for 15 minutes to degrade cellular proteins. The final samples were diluted 1:100 in 1X PBS. To adjust sample loading for equalization, the total protein concentration was determined for each strain. 1 mL of bacterial culture was treated identically as mentioned above for cell lysis. 30 μL of proteinase inhibitor was mixed thoroughly with 180 μL of the lysate to prevent cellular proteins from further degradation. Protein measurement was then performed using Pierce BCA Protein Assay Kit, and the loading volume of the immunoblot samples were readjusted according to their total protein concentration. A vacuum slot-blot device was used to suction samples onto a nitrocellulose membrane. Samples were loaded alongside purified Type 6A capsular polysaccharide standards from Merck. Each sample well was washed with 250 μL 1X PBS. The membrane was blocked with 5% milk in 0.1% PBS-Tween-20 for 30 minutes, washed, and incubated with Type 6A rabbit antiserum (Statens Serum Institute, 1:40000) for 30 minutes at RT. Type 6A antiserum was pre-treated with an un-encapsulated strain to absorb out unspecific binding to non-capsular proteins. The membrane was washed 2 times with 0.1% PBS-Tween-20 for 10 min and incubated with horseradish peroxidase-conjugated IgG goat anti-rabbit antibody (1:5000) for 30 minutes at RT. The membrane was washed 2 times with 0.1% PBS-Tween-20 for 10 min and treated with an ECL substrate for signal amplification. Imaging was performed using the iBright Imaging System, and densitometry was performed using FIJI.

## Depletion of complement

For survival experiments, infant and adult mice were treated with 20 μg or 25 μg of cobra venom factor (CoVF; Quidel Corporation) diluted in 40 μl or 100 μl of dPBS respectively, or dPBS control, at 4 hours prior to and 4 days post-*Spn* infection. For IP *Spn* T4 infection, infant mice were treated with either 20 μg of CoVF or dPBS control 24 hours prior to *Spn* infection.

## RNA isolation and Real-Time PCR

At the indicated time-points, naïve, mock- or *Spn*-infected infant and adult mice were euthanized by $CO_2$ asphyxiation followed by cardiac puncture. To obtain RNA from the URT, the trachea was cannulated and lavaged with 600 μl of RLT lysis buffer (QIAGEN;) containing 1% 2-mercaptoethanol (Sigma). RNA was isolated using a QIAshredder kit (QIAGEN) followed by an RNeasy minikit (QIAGEN) per the manufacturer's protocol. Total RNA was reverse transcribed using a High-Capacity cDNA Reverse Transcription Kit (ThermoFisher). cDNA was purified using a MiniElute PCR Purification Kit (QIAGEN) according to manufacturer's protocol, except cDNA was eluted in 35 μl of UltraPure DNase/RNase-free water (Invitrogen). After elution cDNA was diluted with 80 μl of DNase/RNase-free water. Real-time PCR was performed using 4 μl of cDNA, 0.625 μl of each gene specific primer (10 μM), 12.5 μl of SYBR green (Applied Biosystems) and 7.25 μl of DNase/RNase-free water per sample using a CFX384 Touch Real-Time PCR Detection System (BioRad). Target genes were normalized to respective *Gapdh* levels and data represent fold change according to the ΔΔCt method. The primer sequences for each gene are listed in Box 1.

## Box 1. Primer sequences

| Gene | Organism | Forward and Reverse Sequences 5'→3' |
|---|---|---|
| *Gapdh* | *Mus musculus* | AGGTCGGTGTGAACGGATTTG<br>TGTAGACCATGTAGTTGAGGTCA |
| *Cxcl1* | *Mus musculus* | CTGGGATTCACCTCAAGAACATC<br>CAGGGTCAAGGCAAGCCTC |
| *Cxcl2* | *Mus musculus* | CCACCAACCACCAGGCTAC<br>GCTTCAGGGTCAAGGGCAAA |
| *Lcn2* | *Mus musculus* | ACATTTGTTCCAAGCTCCAGGGC<br>CATGGCGAACTGGTTGTAGTCCG |
| *Cldn8* | *Mus musculus* | GGAATGCCAATCCATCACGC<br>CTCTTTTATCCCCAGGCCCC |
| *Arpc1b* | *Mus musculus* | AGAGTAACCGCATTGTGACCT<br>CGGGCAGCTCGATTGATCC |

## Flow cytometry

Flow cytometry analysis was performed to characterize the cellular composition of nasal tissue. Infant and adult mice were euthanized by $CO_2$ asphyxiation followed by cardiac puncture. To isolate nasal tissue cells, heads were detached and the skin, lower jaw, tongue, eyes, and incisors were removed. The skull was cut laterally, starting behind the ears, and the posterior skull and brain tissues were removed. The remaining tissue was cut into small pieces and digested with collagenase type II (200 U/ml; ThermoFisher) and DNase (15 U/ml; Sigma-Aldrich) in DMEM media (Gibco) for 1 hour and 15 min at 37°C on a rocker. Tubes were shaken vigorously for 30 seconds every half hour during incubation. Cells were passed through a 70 μm filter and centrifuged for 10 min at 500xg at 4°C. Cells were treated with ACK lysing buffer (Lonza) for 5 min to lyse red blood cells, then washed with dPBS and centrifuged for 10 min at 500xg at 4°C. The pellet was re-suspended in dPBS and cells were passed through a 40 μm filter; the suspension was centrifuged for 10 min at 500xg at 4°C. Single nasal cell suspensions were re-suspended in dPBS and stained with a Live/Dead Fixable Green Dead Cell Stain Kit (ThermoFisher) according to the manufacturer's protocol. After Live/Dead staining, cells were washed with dPBS and re-suspended in 50 μl of fluorescence-activated cell sorter (FACS) buffer (dPBS containing 1% BSA and 2 mM EDTA). Cells were stained with a FC receptor-blocking antibody, anti-CD16/32 (BioLegend; clone 93) for 5 minutes at 4°C and then stained with a cocktail of anti-CD45 APC-Cy7 (BD; clone 30-F11), anti-F4/80 PE (BioLegend; clone BM8), anti-CD11b V450 (BD; clone M1/70) and anti-Ly6G PerCp/Cy5.5 (BD; clone 1A8) and anti-Ly6C APC (Biolegend; clone HK1.4) for 30 minutes at 4°C. Cells were washed in FACS buffer, fixed with 4% paraformaldehyde (Affymetrix) for 30 minutes at 4°C and then re-suspended in 50 μl of FACS buffer. Flow cytometry analysis was performed using a LSRII apparatus (Becton Dickinson). Resulting data was analyzed using FlowJo software (Treestar inc., Ashland, OR). Gates were based on Fluorescence-Minus-One (FMO) controls.

## *In vivo* neutrophil depletion

For neutrophil depletion experiments, infant mice were injected intraperitoneally with 200 μg of either rat anti-mouse Gr-1 monoclonal antibody (BioXcell; RB6-8C5; cat. #BE0075) or rat IgG2b isotype control (BioXcell; IgG2b; cat. #BE0090) diluted in 40 μl of dPBS. The antibodies

were administered twice, on the day of *Spn* infection and 48 hours post-infection. Infant mice were euthanized 24 hours after the second dose.

## Blood neutrophil and monocyte counts

To quantify neutrophils and monocytes, blood was harvested through cardiac puncture and collected in an EDTA-coated microtainer tube (BD Biosystems). Samples were run on the Element HT5 Veterinary Hematology Analyzer (Heska) for data acquisition.

## FITC-dextran

To determine barrier permeability, infant and adult mice were IN treated with 3–5 µl of 4 kDa or 2000 kDa FITC-dextran at 20 mg/kg, or PBS control and euthanized by $CO_2$ asphyxiation 1-hour post-treatment. To minimize the potential contribution of FITC-dextran translocation from the gastrointestinal tract, the FITC-dextran was titrated and administered using both an oral and IN route to determine the optimal IN dose. Blood was collected by cardiac puncture, collected in an EDTA-coated microtainer tube (BD Biosystems) and centrifuged for 10 min at 8,000 rpm to obtain plasma. Nasal lavage samples were collected by tracheal lavage using 500 µl of PBS. Plasma and nasal lavage samples were diluted 1:4 or 1:2, respectively, with PBS, and 100µl of each sample was transferred to a black opaque-bottom 96-well plate in duplicate. Readings of relative fluorescence units by a SpectraMax M3 microplate reader (Molecular devices) was determined at 520 nm with excitation at 488 nm. A PBS blank was used as a control to subtract background fluorescence.

## Mouse RNA-Sequencing

Infant (4-day-old) and adult (8-week-old) mice were mock- or *Spn* T23F infected with $1x10^3$ or $1x10^5$ CFU, respectively, and were euthanized 3-days post-infection by $CO_2$ asphyxiation followed by cardiac puncture. Mouse tracheas were lavaged with 200 µl of PBS followed by a second lavage with 600 µl of RLT lysis buffer to collect RNA. RNA was extracted using the QIAshredder kit followed by an RNeasy minikit, as described above. RNA quality was checked using a bioanalyzer, and five samples/group were used for RNA-sequencing. For library preparation, the Illumina TruSeq Stranded mRNA Library Prep Kit (Illumina) was used according to the manufacturer's protocol. A total of 350 ng RNA/sample with 11 cycles for final amplification was performed. Sequencing was performed using Illumina Hi-seq and the raw fastq reads were aligned to mm10 mouse references genome using STAR aligner [53]. Fastq Screen was used to check for any contamination in the samples and Picard RNA-seq Metrics was used to obtain the metrics of all aligned RNA-seq reads. *featureCounts* [54] was used to quantify the gene expression levels. Raw gene counts were used for differential expression analysis. To identify DEGs, *DESeq2* R package [55] was used and DEGs were analyzed using online annotation tool Database for Annotation, Visualization and Integrated Discovery (DAVID) [56,57]. Heat maps were generated using GraphPad Prism 9.0 (GraphPad Software Inc., San Diego, CA). RNA-sequencing data is available in the GEO repository under accession number GSE116604.

## Human study

Details on study design and inclusion criteria were published previously [58,59]. Briefly, non-respiratory tract infection (RTI) nasopharyngeal samples were collected from 112 healthy infants at up to 11 time-points (2 hours after birth, at 24 hours, at 7 and 14 days, and at 1, 2, 3, 4, 6, 9, and 12 months of age). For the study, we selected the final 43 inclusions ($n$ = 286 samples), which were stored in RNA protect Cell Reagent (QIAGEN) at -80˚C. RNA extraction

was performed by the TRIzol/chloroform method, subsequently cleaning samples with the RNAeasy Micro Kit (QIAGEN)(OpenWetWare contributors, 2015). Quality and quantity of RNA was assessed using 2100 Bioanalyzer (Agilent). Further processing of samples was done by Hologic Ltd. (Manchester UK); cRNA was prepared and hybridized onto Affymetrix Clariom S Human Pico arrays.

Bio-informatic quality control was performed as previously described [60]. Following, robust multichip averaging (RMA) background correction and quantile normalization was performed. $Log_2$-transformed gene expression values were used for downstream analyses. Statistical analyses were performed in R version 4.2.1 within R studio version 2023.03.0+386 (Boston, MA). Gene expression data have been deposited at the National Centre for Biotechnology Information GenBank database under accession no. GSE152951. Full patient metadata are available upon request from the Spaarne hospital at WetenschapsBureau@spaarnegasthuis.nl.

### *Spn* serotype 6 staining

Immunostaining was performed on a Leica BondRX automated immunostainer, according to the manufacturer's instructions. In brief, 5 μm thick paraffin sections underwent deparaffinization online followed by epitope retrieval for 60 minutes at 100˚C with Leica Biosystems ER2 solution (pH9, AR9640) and endogenous peroxidase activity blocking with $H_2O_2$. Sections were incubated with primary antibody against *Spn* Serotype 6 (Group 6 rabbit antiserum; Statens Serum Institut) at a 1:1000 dilution for 60 minutes at room temperature. The primary antibody was detected with anti-rabbit HRP-conjugated polymer and 3,3'-diaminobenzidine (DAB) substrate that are provided in the Leica BOND Polymer Refine Detection System (Cat # DS9800). Following counter-staining with hematoxylin, slides were scanned at 40X on a Leica AT2 whole slide scanner (Aperio Image Library v12.0.16, Leica Biosystems) and the image files uploaded to the NYUGSoM's OMERO Plus image data management system (Glencoe Software).

### Statistical analysis

All statistical analyses, excluding the human study, were performed using GraphPad Prism 9.0 (GraphPad Software Inc., San Diego, CA). A Shapiro-Wilk test for normal distribution was performed and the statistical test is noted in figure legends.

### Supporting information

**S1 Fig. A,** Percent survival of adult mice IN infected 1 x $10^5$, 1 x $10^6$ or 1 x $10^7$ CFU of *Spn* T6A at 21 dpi (n = 8–10). Data are collected from one experiment. (TIF)

**S2 Fig. A,** Blood CFU from infant and adult mice intraperitoneally (IP) infected with $10^2$ CFU of *Spn* T6A at 1 dpi (n = 8–13). Statistical significance determined using Mann-Whitney test. ns, not significant. Data represent individual mice with median and are collected from two independent experiments. **B,** Immunoblot of capsule levels from purified *Spn* T6A capsule standard (Std.), *Spn* T6A (WT) and *Spn* T6A CpsE$_{Phe297Ile}$ mutant. **C,** Nasal lavage CFU from infant mice IN infected with either *Spn* T6A (WT) or *Spn* T6A CpsE$_{Phe297Ile}$ mutant at 14 dpi (n = 9–10). Data represent individual mice with median and are collected from 1–2 experiments. (TIF)

**S3 Fig. A,** Nasal lavage CFU from infant mice IN infected with either *Spn* T6A or *Spn* T4 at 3 dpi (n = 11–12). Data represent individual mice with mean ±SEM and are collected from 2

independent experiments. Statistical significance determined using Mann-Whitney test. ns, not significant.
(TIF)

**S4 Fig. A-D,** Infant mice IP treated with IgG isotype control (IgG) or anti-Gr-1 ($\alpha$Gr-1) antibody and IN infected with *Spn* T6A at 3 dpi (n = 9–12). **A**, Number of neutrophils in blood. **B**, Percentage of neutrophils (Live CD45$^+$ CD11b$^+$ Ly6G$^+$) in nasal tissue. Mouse nasal tissue was pooled n = 2–3 per sample. **C,** Percentage of monocytes in blood. **D**, Percentage of monocytes (Live CD45$^+$ CD11b$^+$ Ly6G$^-$ Ly6C$^+$) in nasal tissue. Mouse nasal tissue was pooled n = 2–3. Data represent individual mice with mean ±SEM and are collected from 2–3 independent experiments. Statistical significance determined using unpaired Students *t* test or Mann-Whitney test. **, $p \leq 0.01$; ***, $p \leq 0.001$; ****, $p \leq 0.0001$; ns, not significant.
(TIF)

**S5 Fig. A,** *Spn* T6A molecularly-barcoded library diversity. Histogram indicates the frequency occurrence (Y-axis) of the number of reads per barcode (X-axis). **B**, Number of unique clones and (**C**) proportion of most abundant clone in the blood from infant mice IP infected with *Spn* T6A barcoded library at 1 dpi. Data represent individual mice with median and are collected from one experiment. **D**, Nasal lavage and (**E**) blood CFU from infant mice treated IP with dPBS or cobra venom factor (CoVF) and IN infected with *Spn* T6A barcoded library at 3 dpi (n = 5–6). Data represent individual mice with mean ±SEM and are representative of one experiment. Statistical significance determined using Mann-Whitney test. ns, not significant. **F**, Hill's $N_1$ diversity coefficient in nasal lavage samples collected from septic infant mice that were treated IP with dPBS or cobra venom factor (CoVF) and IN infected with *Spn* T6A barcoded library. Data represent individual mice with mean ±SEM and are collected from two independent experiments. Statistical significance determined using Students *t* test. ns, not significant.
(TIF)

**S1 Data. Source data for main figure graphs in this study.**
(XLSX)

**S2 Data. Source data for supplemental figure graphs in this study.**
(XLSX)

## Acknowledgments

We would like to thank the Applied Bioinformatics Center and the Genome Technology Center at NYU School of Medicine for providing library preparation, sequencing and bioinformatics support. We thank members of the Experimental Pathology Research Laboratory, which is partially supported by the Cancer Center Support Grant P30CA016087 at NYU Langone's Laura and Isaac Perlmutter Cancer Center. We thank Dr. Gregory Putzel and the Microbial Genomics Core Lab for computational support.

## Author Contributions

**Conceptualization:** Kristen L. Lokken-Toyli, Wouter A. A. de Steenhuijsen Piters.

**Data curation:** Kristen L. Lokken-Toyli, Surya D. Aggarwal, Gavyn Chern Wei Bee, Wouter A. A. de Steenhuijsen Piters, Cindy Wu, Kenny Zhi Ming Chen, Cynthia Loomis.

**Formal analysis:** Kristen L. Lokken-Toyli, Surya D. Aggarwal, Gavyn Chern Wei Bee, Wouter A. A. de Steenhuijsen Piters, Cindy Wu, Kenny Zhi Ming Chen, Jeffrey N. Weiser.

**Funding acquisition:** Kristen L. Lokken-Toyli, Jeffrey N. Weiser.

**Investigation:** Kristen L. Lokken-Toyli, Surya D. Aggarwal, Wouter A. A. de Steenhuijsen Piters.

**Methodology:** Kristen L. Lokken-Toyli, Surya D. Aggarwal, Wouter A. A. de Steenhuijsen Piters, Cindy Wu, Cynthia Loomis.

**Project administration:** Jeffrey N. Weiser.

**Resources:** Cynthia Loomis, Jeffrey N. Weiser.

**Software:** Surya D. Aggarwal, Wouter A. A. de Steenhuijsen Piters.

**Supervision:** Debby Bogaert, Jeffrey N. Weiser.

**Validation:** Kristen L. Lokken-Toyli, Wouter A. A. de Steenhuijsen Piters.

**Visualization:** Kristen L. Lokken-Toyli, Wouter A. A. de Steenhuijsen Piters, Cindy Wu, Kenny Zhi Ming Chen, Cynthia Loomis.

**Writing – original draft:** Kristen L. Lokken-Toyli.

**Writing – review & editing:** Kristen L. Lokken-Toyli, Surya D. Aggarwal, Gavyn Chern Wei Bee, Wouter A. A. de Steenhuijsen Piters, Cindy Wu, Kenny Zhi Ming Chen, Cynthia Loomis, Debby Bogaert, Jeffrey N. Weiser.

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
