## [Decision Letter · Decision Letter 0]

11 Dec 2023

Dear Dr. Lokken-Toyli,

Thank you very much for submitting your manuscript "Impaired upper respiratory tract barrier function during postnatal development predisposes to invasive pneumococcal disease" for consideration at PLOS Pathogens. As with all papers reviewed by the journal, your manuscript was reviewed by members of the editorial board and by several independent reviewers. In light of the reviews (below this email), we would like to invite the resubmission of a significantly-revised version that takes into account the reviewers' comments.

Reviewers find the manuscript to be on an important topic and the overall premise of impaired barrier function to be potentially valid explanation for enhanced susceptibility of young animals/infants. However, all reviewers felt that the rigor of the work could be improved with inclusion of key controls, better explanation of methodology/presentation of the data, and some additional data.

We cannot make any decision about publication until we have seen the revised manuscript and your response to the reviewers' comments. Your revised manuscript is also likely to be sent to reviewers for further evaluation.

Sincerely,

Carlos J. Orihuela, PhD

Academic Editor

PLOS Pathogens

Michael Otto

Section Editor

PLOS Pathogens

Kasturi Haldar

Editor-in-Chief

PLOS Pathogens

orcid.org/0000-0001-5065-158X

Michael Malim

Editor-in-Chief

PLOS Pathogens

orcid.org/0000-0002-7699-2064

Reviewers find the manuscript to be on an important topic and the overall premise of impaired barrier function to be potentially valid explanation for enhanced susceptibility of young animals/infants. However, all reviewers felt that the rigor of the work could be improved with inclusion of key controls, better explanation of methodology/presentation of the data, and some additional data.

Reviewer's Responses to Questions

**Part I - Summary**

Reviewer #1: Overall, this manuscript provides convincing evidence that barrier development in the URT is a key factor in susceptibility to pneumococcal infection. There is extensive experimental evidence, including several negative results (which are important controls for this study and should remain included in the paper), that support this hypothesis. I was particularly impressed by the human validation cohort, which recapitulated much of the observations made the murine system. The conclusions were well supported by the experimental data and the manuscript was presented in a clear and logical fashion. The methods were well detailed and easy to follow.

Reviewer #2: In this manuscript, Lokken-Toyli et al. investigate the age-dependent mechanisms by which S. pneumoniae crosses the upper respiratory tract (URT) barrier in infants but not in adults. Using a mouse model, they show that age-dependence correlates with an infection-independent increase in URT barrier permeability and delayed clearance of mucosa-adherent bacteria in young mice. This correlates with upregulated expression of genes encoding tight junction proteins in both adult mice and humans.

Reviewer #3: Lokken-Toyli et al. observe an increased mortality and blood CFU to i.n. S. pneumoniae (T6A) infection in infant (d10) vs adult mice and aim at identifying the underlying reasons for age-dependent infection susceptibility. Notably, this observation is reminiscent to the situation in humans, where infants carry S. p. in their throat and develop occult bloodstream infection prior to invasive infection (meningitis, sepsis) whereas adults develop LRT infection pneumoniae followed by bloodstream infection.

The group has previously published a number of articles on the neonatal S. pn. infection model among others to study host transmission (IAI, 2017) and shown that pneumolysin-induced inflammation drives transmission (CHM, 2017), that susceptibility of infant mice is due to a lack of pneumolysin-induced IL-1a secretion (PLoS Pathogens, 2018), that vaccine induced immunity prevents shedding and transmission (mBio, 2017) and that host transmission is determined by the capsule type (mBio, 2018). In particular, they have shown that pathogen-induced downregulation of tight junction proteins facilitates epithelial translocation (Clarke et al., 2011).

Here, they demonstrate (i) that i.n. and less so i.p. infection with Streptococcus pneumoniae T6A is associated with higher mortality and increased spleen and blood CFUs in infant (d10) mice as compared to adult (8-10w) mice, (ii) that bacterial capsule expression is critical for blood CFU and survival after parenteral infection (i.p.) but not nasal colonization in infant mice, (iii) that complement depletion does not affect survival after i.n. infection of adult or infant mice, (iv) that S.p. T4 exhibits an attenuated phenotype after i.p. infection (survival and blood CFU) and that complement provides protection from systemic T4 i.p. infection (blood CFU, survival), (v) that T6A S. pn. Infection i.n. leads to increased chemokine expression and PMN recruitment (but PMN depletion has no effect on colonization nor blood or spleen CFU or survival), (vi) that deletion of the toxin pneumolysin in T6A S. pn. does not lead to differences in colonization, blood or organs CFU or survival after i.n. infection (with a minor protection by Tlr2 and maybe Il-1R1), (vii) that epithelial translocation represents a major bottleneck independent of complement, (viii) that URT tight junction expression increases with age in mice and man (and FITC dextran translocation decreases in mice), (ix) that tight and adherens junctions expression is reduced in neonates but not adults after S.p. 23F infection but neither 4kDa nor 2000kDa FITC dextran translocation is not increased in neonates infected with T6A, (x) that FITC dextran translocation is enhanced postnatally (until day 14) and infection with T6A at 15-16 days after birth does not lead to mortality (at day 10 it is 39%), FITC dextran clearance from the URT mucosa is delayed up to day 14 after birth, (xi) that i.n. T6A infection leads to relatively lower tissue number as compared to mucosal bacteria in adults as compared to infants and (xii) that T6A leads to close epithelial attachment and transepithelial translocation in infant but not adult mice. Thereby, they show that the enhanced susceptibility in infants following i.n. inoculation has something to do with the early course of the infection and is NOT dependent on bacterial capsule or pneumolysin expression, host complement or PMN recruitment (although enhanced). ORT epithelium translocation generally represents a major population bottleneck and epithelial tight junction expression and luminal clearance is lower in infants than adults and further reduced upon infection. Epithelial attachment and translocation of S. pn. are enhanced in infants.

The research topic is highly relevant. In principle, the data are interesting and novel. In the current text version, the story is hard to follow. Many experiments mainly show negative results for host and bacterial factors. Although consistent with the histological images shown, no function data are provided (and are most likely difficult to generate) that would proof the role of reduced tight junction formation for the enhanced barrier permissiveness for S.p. Other difference might contribute as well.

**Part II – Major Issues: Key Experiments Required for Acceptance**

Reviewer #1: 1. In the section describing the independence of Spn translocation across the URT barrier in infant mice being independent of capsule type- rather that comparing two different strain backgrounds with different capsule types, the more rigorous experiment would be to utilize capsule switching variants that only differ in their capsule locus.

Reviewer #2: While the findings may be important for understanding S. pneumoniae infection in young children, the methods are not detailed enough to be confident of the relevance of the authors' conclusions. In addition, this study reports on correlations or absence of thereof, and the authors conclusions therefore remain speculative. The discussion would benefit from being more concise.

Major points:

Introduction: There is a lack of context to explain what is known, what remains to be investigated and how this study differs from the previous one.

Figure 1: The IN model needs to be better explained: why 103 CFU in infant mice vs. 105 CFU in adults?

Figure 1b, 1c: How are the CFUs for the dead animals estimated? Based on the most infected live animal? If so, they need to be removed from the statistical analysis (estimation and no proof that it's the real data).

Figure 1d: Colonization is not similar on day 1, which could partially explain the increase in bacterial counts in the blood of young mice.

Supplementary Figure 2b: This WB is inconclusive. A reference Spn protein is needed to normalize the amount of capsule. Please show both lines on the same blot to compare the size, or at least with a common marker size.

Figure 2D: A control is missing to know if CovF works: IP infection?

Figure 2E: Were T6A and T4 experiments done in the same experiment? What are the CFUs in nasal lavage for T4?

Figure 2H: Why is the x-axis different from 2e (8 vs 14 days)? Did you follow the animals up to 14 days pi? Would you expect mice with PBS to die at 5 days post infection (compared to no death in Figure 2e)?

Page 10, lines 202 to 205. It can't be concluded that disruption of the URT barrier function allows translocation of Spn independent of serotype. An experiment comparing infant and adult infection with Spn T4 is missing. Complement-independent immune mechanisms may also be involved.

Fig 4e: Tlr2-/- mice are more susceptible to infection than wt. What is the author’s interpretation?

pp13/14 and p28 : Even if already published, the barcode library is not sufficiently described here. How many individual barcodes are there in the library?

Please show the inoculum to compare what is present and what is missing after infection.

Fig 5a: What is the proportion of clones conserved in nasal lavage compared to the inoculum?

Fig 5c: one information is missing: is the most abundant clone the same in nasal lavage and blood or are there different clones?

Fig S4b: What is the inoculum? If there are 103 CFU, how is it possible to have more than 1000 unique clones? How is Hill's N1 calculated and what does it mean? Again, a more detailed description of the method will help.

Fig 6&7: These are nice correlations between the expression of genes encoding tight junction proteins and susceptibility to infection. The fact that fluorescent dyes cross the barrier more efficiently in young mice than in adults (Fig 6d) supports this correlation. Ideally, one would expect to confirm experimentally that this is the main mechanism (compensation of barrier permeability in young, decrease of barrier effect in adults…). However, the authors are cautious in their interpretation ("suggested", "indicated"...).

Fig. 7a: The number of points is very different at different ages. One would expect different results if n were the same for all conditions.

Reviewer #3: 1. The data presented are complex and the logic of their presentation is hard to follow. Restructure the presentation of the findings, maybe move some panels to the supplement, provide clear conclusions from each figure,

2. Comment on the reason for the use of the different S. pn. strains (T4,. T6A, 23F) (or repeat experiments with the similar S.p. strain to allow a comparative analysis) to make it easier to follow the logic of the presented data also for non-S. pn. specialists.

3. Always show in the figure whether infection are i.n. or parenteral (i.p.).

4. Quantify the histological data in figure 7e and f.

5. Please reconcile the findiongs in Fig. 6h and I with the recent publication Clarke et al., 2011.

6. Page 7, “higher blood and spleen CFU despite similar levels of colonization of URT”. The data show a significant difference at day 2 p.i..

7. Please discuss why infant mice (and humans children) might express lower tight junction levels. What could be the benefit of expressing lower levels than in adults.

**Part III – Minor Issues: Editorial and Data Presentation Modifications**

Reviewer #1: Overall, I thought the paper was well-written and studies were well designed. This paper provides substantial value to the field. There were a few parts throughout that needed some clarification, and maybe some points to consider in the discussion.

Line 112: should read, “Streptococcus”

Line 129: I don’t really feel like this is adding anything considering that 10^5 CFU is still considered sub-lethal for adult mice. Is 10^3 CFU still considered sub-lethal for pups?

Line 149: Spn capsule also enhances intracellular survival and vascular endothelial cell translocation, important reference to add (Brissac et al., 2021).

Line 206: This statement conflicts with the section header. The section header states that the phenotype is independent of serotype, but the last sentence directly states it is in fact dependent on serotype.

Line 227: Doesn’t Gr1 also target monocytes? Is there a reason why an antibody targeting specifically neutrophils (i.e., Ly6G) wasn’t used?

Line 241: Phagocytic ROS production has been shown to be essential for bacterial restriction in some infection models (i.e., pulmonary) but not others (skin) using group A Streptococcus (Lei et al., 2017), so it is possible that a similar mechanism may be at play and may be part of why you don’t se

---

## [Decision Letter · Decision Letter 1]

10 Mar 2024

Dear Dr. Lokken-Toyli,

We are pleased to inform you that your manuscript 'Impaired upper respiratory tract barrier function during postnatal development predisposes to invasive pneumococcal disease' has been provisionally accepted for publication in PLOS Pathogens.

Best regards,

Carlos J. Orihuela, PhD

Academic Editor

PLOS Pathogens

Michael Otto

Section Editor

PLOS Pathogens

Michael Malim

Editor-in-Chief

PLOS Pathogens

orcid.org/0000-0002-7699-2064

Plese adjust line 236 and 231 as requested by the reviewer. Otherwise, all concerns and comments were adequately addressed by the authors.

Reviewer Comments (if any, and for reference):

Reviewer's Responses to Questions

**Part I - Summary**

Reviewer #1: The authors addressed my previous concerns and the manuscript is much improved from the previous version.

Reviewer #2: Lokken-Toyli et al. have made significant improvements in the revised version of their manuscript. It includes a more detailed Materials and Methods section, as well as the requested controls: Sup Fig3a, Fig 3A, Fig 3D and Fig 6D.

Reviewer #3: The authors still cannot provide data that functionally link reduced tight junctionm formation with enhanced pathogen translocation. However, I also cannot come up with a possible experimental setup that could demonstrate that link. Also, they exclude many alternative mechanisms adequately and provide human data that suggest that reduced tight junction formation may also account for human neonates/infants. Together, I think the authors did the best they can and have adequately addressed a very important topic of outstanding medical importance.

**Part II – Major Issues: Key Experiments Required for Acceptance**

Reviewer #1: None.

Reviewer #2: The modulation of barrier permeability to definitively prove its role is still missing, but the correlations presented are solid and the interpretations are cautious.

Reviewer #3: see above

**Part III – Minor Issues: Editorial and Data Presentation Modifications**

Reviewer #1: (No Response)

Reviewer #2: Minor

For the anti-Gr-1 antibody, please add on line 236 that it also depletes monocytes, even though you then use CCR2-deficient mice to account for a possible effect of monocytes.

Line 231: "The increased expression of these genes was not dependent on whether the infant mice were septic (Fig. 4a-c)." This is not shown by a statistical analysis because the septic and non-septic points are mixed. Please add this statistical analysis in the Supplement.

Reviewer #3: none

PLOS authors have the option to publish the peer review history of their article (what does this mean?). If published, this will include your full peer review and any attached files.

Reviewer #1: No

Reviewer #2: No

Reviewer #3: No

---

## [Editor Report · Acceptance letter]

19 Apr 2024

Dear Dr. Lokken-Toyli,

We are delighted to inform you that your manuscript, "Impaired upper respiratory tract barrier function during postnatal development predisposes to invasive pneumococcal disease," has been formally accepted for publication in PLOS Pathogens.

Best regards,

Michael Malim

Editor-in-Chief

PLOS Pathogens

orcid.org/0000-0002-7699-2064